# Beyond Prioritized Replay: Sampling States in Model-Based Reinforcement Learning via Simulated Priorities

## Abstract

Prioritized Experience Replay (ER) has been empirically shown to improve sample efficiency across many domains and attracted great attention; however, there is little theoretical understanding of why such prioritized sampling helps and its limitations. In this work, we take a deep look at the prioritized ER. In a supervised learning setting, we show the equivalence between the error-based prioritized sampling method for mean squared error and uniform sampling for cubic power loss. We then provide theoretical insight into why it improves convergence rate upon uniform sampling during early learning. Based on the insight, we further point out two limitations of the prioritized ER method: 1) outdated priorities and 2) insufficient coverage of the sample space. To mitigate the limitations, we propose our model-based stochastic gradient Langevin dynamics sampling method. We show that our method does provide states distributed close to an ideal prioritized sampling distribution estimated by the brute-force method, which does not suffer from the two limitations. We conduct experiments on both discrete and continuous control problems to show our approach's efficacy and examine the practical implication of our method in an autonomous driving application.

## 1 Introduction

Experience Replay (ER) (Lin, 1992) has been a popular method for training large-scale modern Reinforcement Learning (RL) systems (Degris et al., 2012; Adam & Busoniu, 2012; Mnih et al., 2015a; Hessel et al., 2018; François-Lavet et al., 2018). In ER, visited experiences are stored in a buffer, and at each time step, a mini-batch of experiences is *uniformly* sampled to update the training parameters in the value or policy function. Such a method is empirically shown to effectively stabilize the training and improve the sample efficiency of deep RL algorithms. Several follow-up works propose different variants to improve upon it (Schaul et al., 2016; Andrychowicz et al., 2017; Oh et al., 2018; de Bruin et al., 2018; Horgan et al., 2018; Zha et al., 2019; Novati & Koumoutsakos, 2019; Sun et al., 2020). The most relevant one to our work is prioritized ER (Schaul et al., 2016), which attempts to improve the vanilla ER method by sampling those visited experiences proportional to their absolute Temporal Difference (TD) errors. Empirically, it can significantly improve sample efficiency upon vanilla ER on many tested domains.

To gain an intuitive understanding of why the prioritized ER method works, one may recall Model-Based RL (MBRL) methods (Kaelbling et al., 1996; Bertsekas, 2009; Sutton & Barto, 2018). ER can be thought of as an instance of a classical model-based RL architecture—Dyna (Sutton, 1991), using a (limited) non-parametric model given by the buffer (van Seijen & Sutton, 2015; van Hasselt et al., 2019). A Dyna agent uses real experience to update its policy as well as its reward and dynamics model. In-between taking actions, the agent can get hypothetical experiences from the model to further improve the policy. Existing works show that smart ways of sampling those experiences can further improve sample efficiency of a MBRL agent (Sutton et al., 2008; Gu et al., 2016; Goyal et al., 2019; Holland et al., 2018; Pan et al., 2018; Corneil et al., 2018; Janner et al., 2019; Chelu et al., 2020). Particularly, prioritized sweeping (Moore & Atkeson, 1993) improves upon vanilla Dyna. The idea behind prioritized sweeping is quite intuitive: we should give high priority to states whose absolute TD errors are large because they are likely to cause the most change in value estimates.

Hence, applying TD error-based prioritized sampling to ER is a natural idea in a model-free RL setting.

This work provides a theoretical insight into the prioritized ER's advantage and points out its two drawbacks: outdated priorities and insufficient sample space coverage, which may significantly weaken its efficacy. To mitigate the two issues, we propose to leverage the flexibility of using an environment model to acquire hypothetical experiences by simulating priorities. Specifically, we bring in the Stochastic Gradient Langevin Dynamics (SGLD) sampling method to acquire states. We demonstrate that the hypothetical experiences generated by our method are distributed closer to the desired TD error-based sampling distribution, which does not suffer from the two drawbacks. Finally, we demonstrate the utility of our method on various benchmark discrete and continuous control domains and an autonomous driving application.

## 2 BACKGROUND

In this section, we firstly review basic concepts in RL. Then we briefly introduce the prioritized ER method, which will be examined in-depth in the next section. We conclude this section by discussing a classic MBRL architecture called Dyna (Sutton, 1991) and its recent variants, which are most relevant to our work.

**Basic notations.** We consider a discounted Markov Decision Process (MDP) framework (Szepesvári, 2010). A MDP can be denoted as a tuple $(\mathcal{S}, \mathcal{A}, \mathbb{P}, R, \gamma)$ including state space $\mathcal{S}$, action space $\mathcal{A}$, probability transition kernel $\mathbb{P}$, reward function $R$, and discount rate $\gamma \in [0, 1]$. At each environment time step $t$, an RL agent observes a state $s_t \in \mathcal{S}$, takes an action $a_t \in \mathcal{A}$, and moves to the next state $s_{t+1} \sim \mathbb{P}(\cdot|s_t, a_t)$, and receives a scalar reward signal $r_{t+1} = R(s_t, a_t, s_{t+1})$. A policy is a mapping $\pi : \mathcal{S} \times \mathcal{A} \to [0, 1]$ that determines the probability of choosing an action at a given state.

A popular algorithm to find an optimal policy is Q-learning (Watkins & Dayan, 1992). With function approximation, parameterized action-values $Q_\theta$ are updated using $\theta = \theta + \alpha\delta_t\nabla_\theta Q_\theta(s_t, a_t)$ for stepsize $\alpha > 0$ with TD-error $\delta_t \stackrel{\text{def}}{=} r_{t+1} + \gamma\max_{a' \in \mathcal{A}} Q_\theta(s_{t+1}, a') - Q_\theta(s_t, a_t)$. The policy is defined by acting greedily w.r.t. these action-values.

**ER methods.** ER is critical when using neural networks to estimate $Q_\theta$, as used in DQN (Mnih et al., 2015b), both to stabilize and speed up learning. ER method uniformly samples a mini-batch of experiences from those visited ones in the form of $(s_t, a_t, s_{t+1}, r_{t+1})$ to update neural network parameters. Prioritized ER (Schaul et al., 2016) improves upon it by prioritized sampling experiences, where the probability of sampling a certain experience is proportional to its TD error magnitude, i.e., $p(s_t, a_t, s_{t+1}, r_{t+1}) \propto |\delta_t|$. However, the underlying theoretical mechanism behind this method is still not well understood.

**MBRL.** With a model, an agent has more flexibility to sample hypothetical experiences. We consider a one-step model which maps a state-action pair to its possible next state and reward: $\mathcal{P} : \mathcal{S} \times \mathcal{A} \mapsto \mathcal{S} \times \mathbb{R}$. We build on the Dyna formalism (Sutton, 1991) for MBRL, and more specifically, the recently proposed HC-Dyna (Pan et al., 2019) as shown in Algorithm 1. HC-Dyna provides a special approach to *Search-Control* (SC)—the mechanism of generating states or state-action pairs from which to query the model to get the next states and rewards. HC-Dyna's search-control mechanism generates states by Hill Climbing (HC) on some criterion function $h(\cdot)$. The term HC is used for generality as the vanilla gradient ascent is modified to resolve certain challenges (Pan et al., 2019).

The algorithmic framework maintains two buffers: the conventional ER buffer storing experiences (i.e., an experience/transition has the form of $(s_t, a_t, s_{t+1}, r_{t+1})$) and a *search-control queue* storing the states acquired by search-control mechanisms. At each time step $t$, a real experience $(s_t, a_t, s_{t+1}, r_{t+1})$ is collected and stored into ER buffer. Then the HC search-control process starts to collect states and store them into the search-control queue. A hypothetical experience is obtained by first selecting a state $s$ from the search-control queue, then selecting an action $a$ according to the current policy, and then querying the model to get the next state $s'$ and reward $r$ to form an experience $(s, a, s', r)$. These hypothetical transitions are combined with real experiences into a single mini-batch to update the training parameters. The $n$ updates, performed before taking the next action, are called *planning updates* (Sutton & Barto, 2018), as they improve the value/policy by using

---

**Algorithm 1** HC-Dyna: Generic framework

---

**Input:** Hill Climbing (HC) criterion function $h : \mathcal{S} \mapsto \mathbb{R}$, batch-size $b$; Initialize empty search-control queue $B_{sc}$; empty ER buffer $B_{er}$; initialize policy and model $\mathcal{P}$; HC stepsize $\alpha_h$; mini-batch size $b$; environment $\mathcal{P}$; mixing rate $\rho$ decides the proportion of hypothetical experiences in a mini-batch.

**for** $t = 1, 2, \ldots$ **do**
    Add $(s_t, a_t, s_{t+1}, r_{t+1})$ to $B_{er}$
    **while** within some budget time steps **do**
        $s \leftarrow s + \alpha_h \nabla_s h(s)$ // HC for search-control
        Add $s$ into $B_{sc}$
    // $n$ planning updates/steps
    **for** $n$ times **do**
        $B \leftarrow \emptyset$ // initialize an empty mini-batch $B$
        **for** $b\rho$ times **do**
            Sample $s \sim B_{sc}$, on-policy action $a$
            Sample $s', r \sim \mathcal{P}(s, a)$
            $B \leftarrow (s, a, s', r)$
        Sample $b(1 - \rho)$ experiences from $B_{er}$, add to $B$
        Update policy/value on mixed mini-batch $B$

---

a model. The choice of pairing states with on-policy actions to form hypothetical experiences has been reported to be beneficial (Gu et al., 2016; Pan et al., 2018; Janner et al., 2019).

Two instances have been proposed for $h(\cdot)$: the value function $v(s)$ (Pan et al., 2019) and the gradient magnitude $||\nabla_s v(s)||$ (Pan et al., 2020). The former is used as a measure of the utility of a state: doing HC on the learned value function should find high-value states without being constrained by the physical environment dynamics. The latter is considered as a measure of the value approximation difficulty, then doing HC provides additional states whose values are difficult to learn. The two suffer from several issues as we discuss in the Appendix A.1. This paper will introduce a HC search-control method motivated by overcoming the limitations of the prioritized ER.

## 3 A DEEPER LOOK AT ERROR-BASED PRIORITIZED SAMPLING

In this section, we provide theoretical motivation for error-based prioritized sampling by showing its equivalence to optimizing a cubic power objective with uniform sampling in a supervised learning setting. We prove that optimizing the cubic objective provides a faster convergence rate during early learning. Based on the insight, we discuss two limitations of the prioritized ER: 1) outdated priorities and 2) insufficient coverage of the sample space. We then empirically study the such limitations.

### 3.1 THEORETICAL INSIGHT INTO ERROR-BASED PRIORITIZED SAMPLING

In the $l_2$ regression, we minimize the mean squared error $\min_\theta \frac{1}{2n} \sum_{i=1}^{n} (f_\theta(x_i) - y_i)^2$, for training set $\mathcal{T} = \{(x_i, y_i)\}_{i=1}^{n}$ and function approximator $f_\theta$, such as a neural network. In error-based prioritized sampling, we define the priority of a sample $(x, y) \in \mathcal{T}$ as $|f_\theta(x) - y|$; the probability of drawing a sample $(x, y) \in \mathcal{T}$ is typically $q(x, y; \theta) \propto |f_\theta(x) - y|$. We employ the following form to compute the probability of a point $(x, y) \in \mathcal{T}$:

$$q(x, y; \theta) \stackrel{\text{def}}{=} \frac{|f_\theta(x) - y|}{\sum_{i=1}^{n} |f_\theta(x_i) - y_i|}. \tag{1}$$

We can show an equivalence between the gradients of the squared objective with this prioritization and the cubic power objective $\frac{1}{3n} \sum_{i=1}^{n} |f_\theta(x_i) - y_i|^3$ in the Theorem 1 below. See Appendix A.3 for the proof.

**Theorem 1.** *For a constant $c$ determined by $\theta, \mathcal{T}$, we have*

$$c\mathbb{E}_{(x,y) \sim q(x,y;\theta)}[\nabla_\theta (1/2)(f_\theta(x) - y)^2] = \mathbb{E}_{(x,y) \sim uniform(\mathcal{T})}[\nabla_\theta (1/3)|f_\theta(x) - y|^3].$$

We empirically verify this equivalence in the Appendix A.7. This simple theorem provides an intuitive reason for why prioritized sampling can help improve sample efficiency: the gradient direction of

the cubic function is sharper than that of the square function when the error is relatively large (Figure 8). We refer readers to the work by Fujimoto et al. (2020) regarding more discussions about the equivalence between prioritized sampling and of uniform sampling. Our Theorem 2 further proves that optimizing the cubic power objective by gradient descent has faster convergence rate than the squared objective, and this provides a solid motivation for using error-based prioritized sampling. See Appendix A.4 for a more detailed version of the theorem (which includes an additional conclusion) and its proof and empirical simulations.

**Theorem 2** (Fast early learning, concise version)**.** *Let $n$ be a positive integer (i.e., the number of training samples). Let $x_t, \tilde{x}_t \in \mathbb{R}^n$ be the target estimates of all samples at time $t, t \geq 0$. Let $x_t(i)(i \in [n], [n] \overset{\text{def}}{=} \{1, 2, ..., n\})$ denote the $i$th element in the vector. We define the objectives:*

$$\ell_2(x, y) \overset{\text{def}}{=} \frac{1}{2} \cdot \sum_{i=1}^n (x(i) - y(i))^2, \quad \ell_3(x, y) \overset{\text{def}}{=} \frac{1}{3} \cdot \sum_{i=1}^n |x(i) - y(i)|^3.$$

*The total absolute prediction errors:*

$$\delta_t \overset{\text{def}}{=} \sum_{i=1}^n \delta_t(i) = \sum_{i=1}^n |x_t(i) - y(i)|, \quad \tilde{\delta}_t \overset{\text{def}}{=} \sum_{i=1}^n \tilde{\delta}_t(i) = \sum_{i=1}^n |\tilde{x}_t(i) - y(i)|,$$

*where $y(i) \in \mathbb{R}$ is the training target for the $i$th training sample. Let $\{x_t\}_{t \geq 0}$ and $\{\tilde{x}_t\}_{t \geq 0}$ be generated by using $\ell_2, \ell_3$ objectives respectively. That is, $\forall i \in [n]$,*

$$\frac{dx_t(i)}{dt} = -\eta \cdot \frac{d\ell_2(x_t, y)}{dx_t(i)}, \quad \frac{d\tilde{x}_t(i)}{dt} = -\eta \cdot \frac{d\ell_3(\tilde{x}_t, y)}{d\tilde{x}_t(i)}.$$

*Given any $0 < \epsilon \leq \delta_0 = \sum_{i=1}^n \delta_0(i)$, define the following hitting time,*

$$t_\epsilon \overset{\text{def}}{=} \min_t \{t \geq 0 : \delta_t \leq \epsilon\}, \quad \tilde{t}_\epsilon \overset{\text{def}}{=} \min_t \{t \geq 0 : \tilde{\delta}_t \leq \epsilon\}.$$

*Assume the same initialization $x_0 = \tilde{x}_0$. **We have the following conclusion**.*
*If there exists $\delta_0 \in \mathbb{R}$ and $0 < \epsilon \leq \delta_0$ such that*

$$\frac{1}{n} \cdot \sum_{i=1}^n \frac{1}{\delta_0(i)} \leq \frac{\log(\delta_0/\epsilon)}{\frac{\delta_0}{\epsilon} - 1}, \tag{2}$$

*then we have $t_\epsilon \geq \tilde{t}_\epsilon$, which means gradient descent using the cubic loss function will achieve the total absolute error threshold $\epsilon$ faster than using the squared objective function.*

This theorem illustrates that when the total loss of all training examples is greater than some threshold, cubic power learns faster. For example, let the number of samples $n = 1000$, and each sample has initial loss $\delta_0(i) = 2$. Then $\delta_0 = 2000$. Setting $\epsilon = 570$(i.e., $\epsilon(i) \approx 0.57$) satisfies the inequality (2). This implies using the cubic objective is faster when reducing the total loss from 2000 to 570. Though it is not our focus here to investigate the practical utility of the high power objectives, we include some empirical results and discuss the practical utilities of such objectives in Appendix A.6.

## 3.2 LIMITATIONS OF THE PRIORITIZED ER

Inspired by the above theorems, we now discuss two drawbacks of prioritized sampling: **outdated priorities** and **insufficient sample space coverage**. Then we empirically examine their importance and effects in the next section.

The above two theorems show that the advantage of prioritized sampling comes from the faster convergence rate of cubic power objective during early learning. By Theorem 1, such advantage requires to update the priorities of *all training samples* by using the *updated training parameters $\theta$* at each time step. In RL, however, at the each time step $t$, the original prioritized ER method only updates the priorities of those experiences from the sampled mini-batch, leaving the priorities of the rest of experiences unchanged (Schaul et al., 2016). We call this limitation **outdated priorities**. It is typically infeasible to update the priorities of all experiences at each time step.

In fact, in RL, "*all training samples*" in RL are restricted to those visited experiences in the ER buffer, which may only contain a small subset of the whole state space, making the estimate of the prioritized

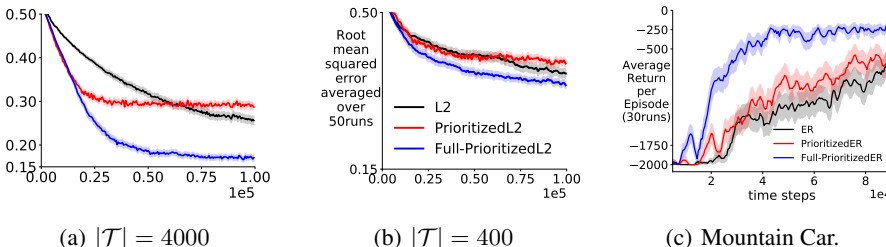

(a) $|\mathcal{T}| = 4000$      (b) $|\mathcal{T}| = 400$      (c) Mountain Car.

Figure 1: Comparing **L2 (black)**, **PrioritizedL2 (red)**, and **Full-PrioritizedL2 (blue)** in terms of testing RMSE v.s. number of mini-batch updates. (a)(b) show the results trained on a large and small training set, respectively. (c) shows the result of a corresponding RL experiment on mountain car domain. We compare episodic return v.s. environment time steps for **ER (black)**, **PrioritizedER (red)**, and **Full-PrioritizedER (blue)**. Results are averaged over 50 random seeds on (a), (b) and 30 on (c). The shade indicates standard error.

sampling distribution inaccurate. There can be many reasons for the small coverage: the exploration is difficult, the state space is huge, or the memory resource of the buffer is quite limited, etc. We call this issue **insufficient sample space coverage**, which is also noted by Fedus et al. (2020).

Note that *insufficient sample space coverage* should not be considered equivalent to off-policy distribution issue. The latter refers to some old experiences in the ER buffer may be unlikely to appear under the current policy (Novati & Koumoutsakos, 2019; Zha et al., 2019; Sun et al., 2020; Oh et al., 2021). In contrast, the issue of insufficient sample space coverage can raise naturally. For example, the state space is large and an agent is only able to visit a small subset of the state space during early learning stage. We visualize the state space coverage issue on a RL domain in Section 4.

### 3.3 Negative Effects of the Limitations

In this section, we empirically show that the outdated priorities and insufficient sample space coverage significantly blur the advantage of the prioritized sampling method.

**Experiment setup**. We conduct experiments on a supervised learning task. We generate a training set $\mathcal{T}$ by uniformly sampling $x \in [-2, 2]$ and adding zero-mean Gaussian noise with standard deviation $\sigma = 0.5$ to the target $f_{\sin}(x)$ values. Define $f_{\sin}(x) \stackrel{\text{def}}{=} \sin(8\pi x)$ if $x \in [-2, 0)$ and $f_{\sin}(x) = \sin(\pi x)$ if $x \in [0, 2]$. The testing set contains 1k samples where the targets are not noise-contaminated. Previous work (Pan et al., 2020) show that the high frequency region $[-2, 0]$ usually takes long time to learn. Hence we expect error-based prioritized sampling to make a clear difference in terms of sample efficiency on this dataset. We use $32 \times 32$ tanh layers neural network for all algorithms. We refer to Appendix A.8 for missing details and A.7 for additional experiments.

**Naming of algorithms**. **L2**: the $l_2$ regression with uniformly sampling from $\mathcal{T}$. **Full-PrioritizedL2**: the $l_2$ regression with prioritized sampling according to the distribution defined in (1), the priorities of *all samples* in the training set are updated after each mini-batch update. **PrioritizedL2**: the only difference with **Full-PrioritizedL2** is that *only* the priorities of those training examples sampled in the mini-batch are updated at each iteration, the rest of the training samples use the original priorities. This resembles the approach taken by the prioritized ER in RL (Schaul et al., 2016). We show the learning curves in Figure 1.

**Outdated priorities.** Figure 1 (a) shows that PrioritizedL2 without updating all priorities can be significantly worse than Full-PrioritizedL2. Correspondingly, we further verify this phenomenon on the classical Mountain Car domain (Brockman et al., 2016). Figure 1(c) shows the evaluation learning curves of different DQN variants in an RL setting. We use a small $16 \times 16$ ReLu NN as the $Q$-function, which should highlight the issue of priority updating: every mini-batch update potentially perturbs the values of many other states. Hence many experiences in the ER buffer have the wrong priorities. Full-PrioritizedER does perform significantly better.

**Sample space coverage.** To check the effect of insufficient sample space coverage, we examine how the relative performances of L2 and Full-PrioritizedL2 change when we train them on a smaller training dataset with only 400 examples as shown in Figure 1(b). The small training set has a small coverage of the sample space. Unsurprisingly, using a small training set makes all algorithms perform worse; however, *it significantly narrows the gap between Full-PrioritizedL2 and L2*. This indicates that prioritized sampling needs sufficient samples across the sample space to estimate the prioritized sampling distribution reasonably accurate. We further verify the sample space coverage issue in prioritized ER on a RL problem in the next section.

## 4 ADDRESSING THE LIMITATIONS

In this section, we propose a Stochastic Gradient Langevin Dynamics (SGLD) sampling method to mitigate the limitations of the prioritized ER method mentioned in the above section. Then we empirically examine our sampling distribution. We also describe how our sampling method is used for the search-control component in Dyna.

### 4.1 SAMPLING METHOD

**SGLD sampling method.** Let $v^\pi(\cdot; \theta) : \mathcal{S} \mapsto \mathbb{R}$ be a differentiable value function under policy $\pi$ parameterized by $\theta$. For $s \in \mathcal{S}$, define $y(s) \stackrel{\text{def}}{=} \mathbb{E}_{r,s' \sim \mathcal{P}^\pi(s',r|s)}[r + \gamma v^\pi(s'; \theta)]$, and denote the TD error as $\delta(s, y; \theta_t) \stackrel{\text{def}}{=} y(s) - v(s; \theta_t)$. Given some initial state $s_0 \in \mathcal{S}$, let the state sequence $\{s_i\}$ be the one generated by updating rule $s_{i+1} \leftarrow s_i + \alpha_h \nabla_s \log |\delta(s_i, y(s_i); \theta_t)| + X_i$, where $\alpha_h$ is a stepsize and $X_i$ is a Gaussian random variable with some constant variance.[1] Then $\{s_i\}$ converges to the distribution $p(s) \propto |\delta(s, y(s))|$ as $i \to \infty$. The proof is a direct consequence of the convergent behavior of Langevin dynamics stochastic differential equation (SDE) (Roberts, 1996; Welling & Teh, 2011; Zhang et al., 2017). We include a brief background knowledge in Appendix A.2.

It should be noted that, this sampling method enables us to acquire states *1) whose absolute TD errors are estimated by using current parameter $\theta_t$ and 2) that are not restricted to those visited ones*. We empirically verify the two points in Section 4.2.

**Implementation**. In practice, we can compute the state value estimate by $v(s) = \max_a Q(s, a; \theta_t)$ as suggested by Pan et al. (2019). In the case that a true environment model is not available, we compute an estimate $\hat{y}(s)$ of $y(s)$ by a learned model. Then at each time step $t$, states approximately following the distribution $p(s) \propto |\delta(s, y(s))|$ can be generated by

$$s \leftarrow s + \alpha_h \nabla_s \log |\hat{y}(s) - \max_a Q(s, a; \theta_t)| + X, \tag{3}$$

where $X$ is a Gaussian random variable with zero-mean and some small variance. Observing that $\alpha_h$ is small, we consider $\hat{y}(s)$ as a constant given a state $s$ without backpropagating through it. Though this updating rule introduces bias due to the usage of a learned model, fortunately, the difference between the sampling distribution acquired by the true model and the learned model can be upper bounded as we show in Theorem 3 in Appendix A.5.

**Algorithmic details.** We present our algorithm called **Dyna-TD** in the Algorithm 3 in Appendix A.8. Our algorithm follows Algorithm 1. Particularly, we choose the function $h(s) \stackrel{\text{def}}{=} \log |\hat{y}(s) - \max_a Q(s, a; \theta_t)|$ for HC search-control process, i.e., run the updating rule 3 to generate states.

### 4.2 EMPIRICAL VERIFICATION OF TD ERROR-BASED SAMPLING METHOD

We visualize the distribution of the sampled states by our method and those from the buffer of the prioritized ER, verifying that our sampled states have an obviously larger coverage of the state space. We then empirically verify that our sampling distribution is closer to a brute-force calculated prioritized sampling distribution—which does not suffer from the two limitations—than the prioritized ER method. Finally, we discuss concerns regarding computational cost. Please see Appendix A.8 for any missing details.

**Large sample space coverage**. During early learning, we visualize 2k states sampled from 1) DQN's buffer trained by prioritized ER and 2) our algorithm Dyna-TD's Search-Control (SC) queue on the continuous state GridWorld (Figure 2(a)). Figure 2 (b-c) visualize state distributions with different sampling methods via heatmap. Darker color indicates higher density. (b)(c) show that DQN's ER buffer, no matter with or without prioritized sampling, does not cover well the top-left part and the right half part on the GridWorld. In contrast, Figure 2 (d) shows that states from our SC queue are more diversely distributed on the square. These visualizations verify that our sampled states cover better the sample space than the prioritized ER does.

**Sampling distribution is close to the ideal one**. We denote our sampling distribution as $p_1(\cdot)$, the one acquired by conventional prioritized ER as $p_2(\cdot)$, and the one computed by thorough priority

---

[1]The stepsize and variance affect the temperature parameter. We treat the two as hyper-parameters.

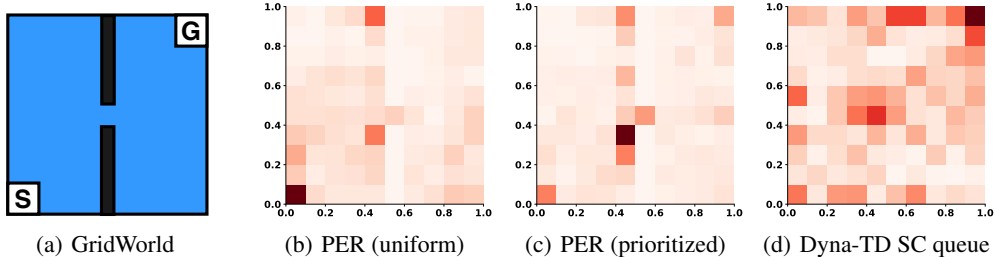

|  (a) GridWorld | (b) PER (uniform) | (c) PER (prioritized) | (d) Dyna-TD SC queue |

Figure 2: (a) shows the GridWorld (Pan et al., 2019). It has $\mathcal{S} = [0, 1]^2, \mathcal{A} = \{up, down, right, left\}$. The agent starts from the left bottom and learn to reach the right top within as few steps as possible. (b) and (c) respectively show the state distributions with uniform and prioritized sampling methods from the ER buffer of prioritized ER. (d) shows the SC queue state distribution of our Dyna-TD.

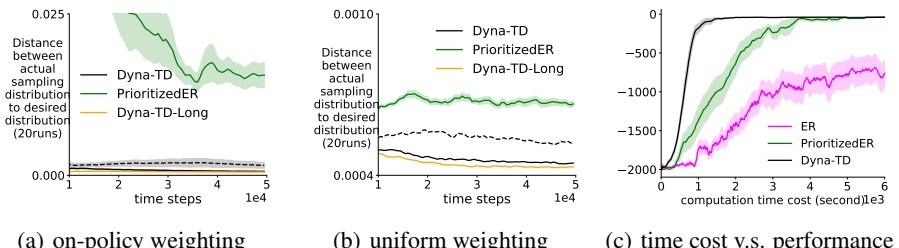

(a) on-policy weighting    (b) uniform weighting    (c) time cost v.s. performance

Figure 3: (a)(b) show the distance change as a function of environment time steps for **Dyna-TD (black)**, **PrioritizedER (forest green)**, and **Dyna-TD-Long (orange)**, with different weighting schemes. The **dashed** line corresponds to our algorithm with an online learned model. The corresponding evaluation learning curve is in the Figure 4(c). (d) shows the policy evaluation performance as a function of running time (in seconds) with **ER(magenta)**. All results are averaged over 20 random seeds. The shade indicates standard error.

updating of enumerating all states in the state space as $p^*(\cdot)$ (this one should be unrealistic in practice and we call it the ideal distribution as it does not suffer from the two limitations we discussed). We visualize how well $p_1(\cdot)$ and $p_2(\cdot)$ can approximate $p^*(\cdot)$ on the GridWorld domain, where the state distributions can be conveniently estimated by discretizing the continuous state GridWorld to a $50 \times 50$ one. We compute the distances of $p_1, p_2$ to $p^*$ by two sensible weighting schemes: 1) on-policy weighting: $\sum_{j=1}^{2500} d^\pi(s_j)|p_i(s_j) - p^*(s_j)|, i \in \{1, 2\}$, where $d^\pi$ is approximated by uniformly sample 3k states from a recency buffer; 2) uniform weighting: $\frac{1}{2500} \sum_{j=1}^{2500} |p_i(s_j) - p^*(s_j)|, i \in \{1, 2\}$.

We plot the distances change when we train our Algorithm 3 and the prioritized ER in Figure 3(a)(b). They show that the HC procedure in our algorithm Dyna-TD, either with a true or an online learned model, produces a state distribution with significantly closer distance to the desired sampling distribution $p^*$ than PrioritizedER under both weighting schemes. In contrast, the state distribution acquired from PrioritizedER, which suffers from the two limitations, is far away from $p^*$. It should also be noted that we include Dyna-TD-Long, which runs a large number of HC steps so that its corresponding sampling distribution should be closer to stationary distribution. However, there is only tiny difference between the regular Dyna-TD and Dyna-TD-Long, implying that one can save computational cost by running fewer HC steps.

**Computational cost.** Let the mini-batch size be $b$, and the number of HC steps be $k_{HC}$. If we assume one mini-batch update takes $\mathcal{O}(c)$, then the time cost of our sampling is $\mathcal{O}(ck_{HC}/b)$, which is reasonable. On the GridWorld, Figure 3(c) shows that given the same time budget, our algorithm achieves better performance. This makes the additional time spent on search-control worth it.

## 5 EXPERIMENTS

In this section, we design experiments to answer the following questions. (1) By mitigating the limitations of conventional prioritized ER method, can Dyna-TD outperform the prioritized ER under various planning budgets in different environments? (2) Can Dyna-TD outperforms the existing Dyna variants? (3) How effective is Dyna-TD under an online learned model, particularly for more realistic applications where actions are continuous, or input dimensionality is higher?

**Baselines.** **ER** is DQN with a regular ER buffer without prioritized sampling. **PrioritizedER** is the one by Schaul et al. (2016), which has the drawbacks as discussed in our paper. **Dyna-Value** (Pan

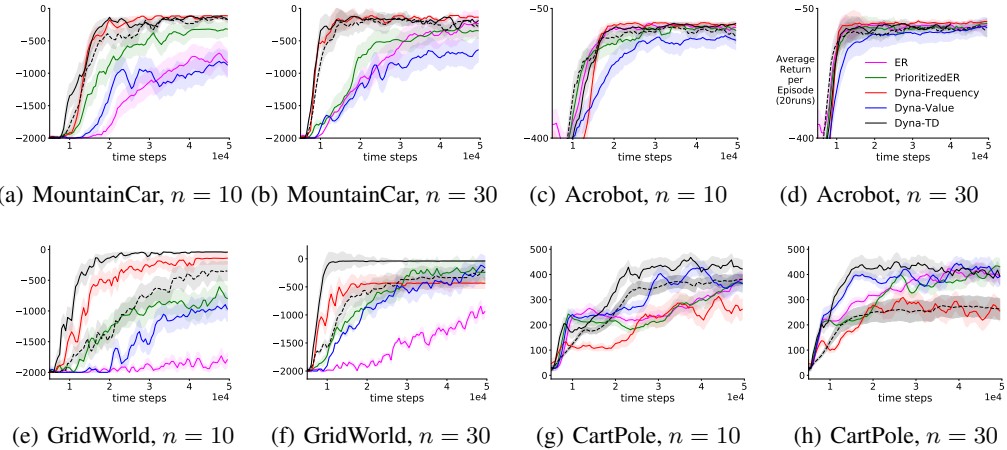

(a) MountainCar, $n = 10$ (b) MountainCar, $n = 30$ (c) Acrobot, $n = 10$ (d) Acrobot, $n = 30$

(e) GridWorld, $n = 10$ (f) GridWorld, $n = 30$ (g) CartPole, $n = 10$ (h) CartPole, $n = 30$

Figure 4: Episodic return v.s. environment time steps. We show evaluation learning curves of **Dyna-TD (black)**, **Dyna-Frequency (red)**, **Dyna-Value (blue)**, **PrioritizedER (forest green)**, and **ER(magenta)** with planning updates $n = 10, 30$. The **dashed** line denotes Dyna-TD with an online learned model. All results are averaged over 20 random seeds after smoothing over a window of size 30. The shade indicates standard error.

et al., 2019) is the Dyna variant which performs HC on the learned value function to acquire states to populate the SC queue. **Dyna-Frequency** (Pan et al., 2020) is the Dyna variant which performs HC on the norm of the gradient of the value function to acquire states to populate the SC queue. For fair comparison, at each environment time step, we stochastically sample the same number of mini-batches to train those model-free baselines as the number of planning updates in Dyna variants. We are able to fix the same HC hyper-parameter setting across all environments. Please see Appendix A.8 for any missing details.

**Performances on benchmarks.** Figure 4 shows the performances of different algorithms on MountainCar, Acrobot, GridWorld (Figure 2(a)), and CartPole. On these small domains, we focus on studying our sampling distribution and hence we need to isolate the effect of model errors (by using a true environment model), though we include our algorithm Dyna-TD with an online learned model for curiosity. We have the following observations. First, our algorithm Dyna-TD consistently outperforms PrioritizedER across domains and planning updates. In contrast, the PrioritizedER may not even outperform regular ER, as occurred in the previous supervised learning experiment. Second, Dyna-TD's performance significantly improves and even outperforms other Dyna variants when increasing the planning budget (i.e., planning updates $n$) from 10 to 30. This validates the utility of those additional hypothetical experiences acquired by our sampling method. In contrast, both ER and PrioritizedER show limited gain when increasing the planning budget (i.e., number of mini-batch updates), which implies the limited utility of those visited experiences. Third, Dyna-Value/Frequency frequently converge to a sub-optimal policy when using a large number of planning updates, while Dyna-TD always finds a better one. It may be that the two Dyna variants frequently generate high-value/frequency states whose TD errors are low, which wastes samples and leads to serious distribution bias. Dyna-Frequency additionally suffers from explosive or zero gradients, and hence is sensitive to hyper-parameters (Pan et al., 2020), which may explain its inconsistent performances.

**A demo for continuous control.** We demonstrate that our approach can be applied for Mujoco (Todorov et al., 2012) continuous control problems with an online learned model and still achieve superior performance. We use DDPG (Deep Deterministic Policy Gradient) (Lillicrap et al., 2016; Silver et al., 2014) as an example for use inside our Dyna-TD. Let $\pi_{\theta'} : \mathcal{S} \mapsto \mathcal{A}$ be the actor, then we set the HC function as $h(s) \overset{\text{def}}{=} \log |\hat{y} - Q_\theta(s, \pi_{\theta'}(s))|$ where $\hat{y}$ is the TD target. Figure 5 (a)(b) shows the learning curves of DDPG trained with ER, PrioritizedER, and our Dyna-TD on Hopper and Walker2d respectively. Since other Dyna variants never show an advantage and are not relevant to the purpose of this experiment, we no longer include them. Dyna-TD shows quick improvement as before. This indicates our sampled hypothetical experiences could be helpful for actor-critic algorithms that are known to be prone to local optimums. Additionally, we note again that ER outperforms PrioritizedER, as occurred in the discrete control and supervised learning (PrioritizedL2 is worse than L2) experiments.

**Autonomous driving application.** We study the practical utility of our method in a relatively large autonomous driving application (Leurent, 2018) with an online learned model. We use the roundabout-v0 domain (Figure 6 (a)). The agent learns to go through a roundabout by lane change and longitude

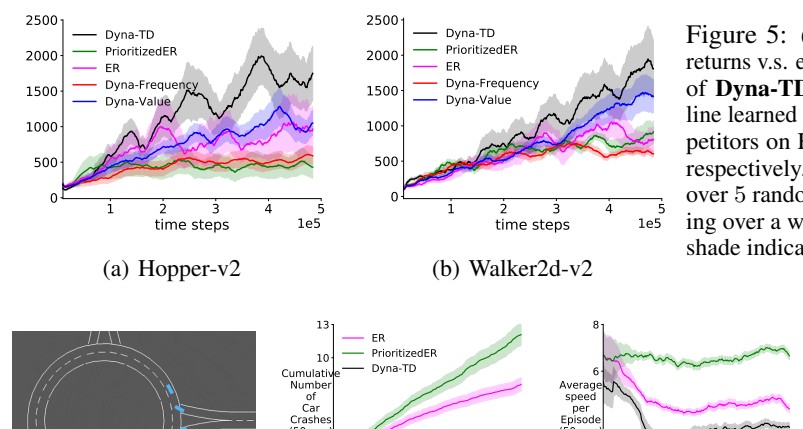

(a) Hopper-v2          (b) Walker2d-v2

Figure 5: (a) (b) show episodic returns v.s. environment time steps of **Dyna-TD (black)** with an on-line learned model, and other competitors on Hopper and Walker2d respectively. Results are averaged over 5 random seeds after smoothing over a window of size 30. The shade indicates standard error.

(a) roundabout    (b) Num of car crashes    (c) Avg. speed    (d) Episodic return

Figure 6: (a) shows the roundabout domain with $\mathcal{S} \subset \mathbb{R}^{90}$. (b) shows crashes v.s. total driving time steps during policy evaluation. (c) shows the average speed per evaluation episode v.s. environment time steps. (d) shows the episodic return v.s. trained environment time steps. We show **Dyna-TD (black)** with an online learned model, **PrioritizedER (forest green)**, and **ER (magenta)**. Results are averaged over 50 random seeds after smoothing over a window of size 30. The shade indicates standard error.

control. The reward is designed such that the car should go through the roundabout as fast as possible without collision. We observe that all algorithms perform similarly when evaluating algorithms by episodic return (Figure 6 (d)). In contrast, there is a significantly lower number of car crashes with the policy learned by our algorithm, as shown in Figure 6(b). Figure 6 (c) suggests that ER and PrioritizedER gain reward mainly due to fast speed which potentially incur more car crashes. The conventional prioritized ER method still incurs many crashes, which may indicate its prioritized sampling distribution does not provide enough crash experiences to learn.

## 6 DISCUSSION

We provide theoretical insight into the error-based prioritized sampling by establishing its equivalence to the uniform sampling for a cubic power objective in a supervised learning setting. Then we identify two drawbacks of prioritized ER: outdated priorities and insufficient sample space coverage. We mitigate the two limitations by SGLD sampling method with empirical verification. Our empirical results on both discrete and continuous control domains show the efficacy of our method.

There are several promising future directions. First, a natural follow-up question is how a model should be learned to benefit our sampling method. Existing results show that learning a model while considering how to use it should make the policy robust to model errors (Farahmand et al., 2017; Farahmand, 2018). Second, one may apply our approach with a model in some latent space (Hamilton et al., 2014; Wahlström et al., 2015; Ha & Schmidhuber, 2018; Hafner et al., 2019; Schrittwieser et al., 2020), which enables our method to scale to large domains. Third, since there are existing works examining how ER is affected by boostrap return (Daley & Amato, 2019), by buffer or mini-batch size (Zhang & Sutton, 2017; Liu & Zou, 2017), and by number of environment steps taken per gradient step (Fu et al., 2019; van Hasselt et al., 2018; Fedus et al., 2020). It is worth studying the theoretical implications of those design choices and their effects on prioritized ER's efficacy.

Last, as our cubic objective explains only one version of the error-based prioritization, efforts should also be made to theoretically interpret other prioritized sampling distributions, such as distribution location or reward-based prioritization (Lambert et al., 2020). It is interesting to explore whether these alternatives can also be formulated as surrogate objectives. Furthermore, a recent work by Fujimoto et al. (2020) establishes an equivalence between various prioritized sampling distributions and uniform sampling for different loss functions, which bears similarities to our Theorem 1. It is interesting to study if those general loss functions have faster convergence rate as shown in our Theorem 2.

## 7 ETHIC STATEMENT

This work is about the methodology of how to sample hypothetical experiences in model-based reinforcement learning efficiently. The potential impact of this work is likely to be further improvement of sample efficiency of reinforcement learning methods, which should be generally beneficial to the reinforcement learning research community. We have not considered specific applications or practical scenarios as the goal of this work. Hence, it does not have any direct ethical consequences.

## 8 REPRODUCIBLE STATEMENT

We commit to ensuring that other researchers with reasonable background knowledge in our area can reproduce our theoretical and empirical results. We provide detailed theoretical derivation for our theorems 1 and 2 in the Appendix, where there is a clear table of contents pointing to different concrete mathematical derivations for these theorems. We also provide sufficient experimental details to reproduce our empirical results in Appendix A.8. We will provide our repository upon acceptance or reviewer's request.

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

## A  APPENDIX

The appendix includes the following contents:

### A.1  BACKGROUND IN DYNA

Dyna integrates model-free and model-based policy updates in an online RL setting (Sutton, 1990). As shown in Algorithm 2, at each time step, a Dyna agent uses the real experience to learn a model and performs a model-free policy update. During the *planning* stage, simulated experiences are acquired from the model to further improve the policy. It should be noted that, in Dyna, the concept of *planning* refers to any computational process which leverages a model to improve policy, according to Sutton & Barto (2018), Chapter 8. The mechanism of generating states or state-action pairs from which to query the model is called *search-control*, which is of critical importance to improving sample efficiency. The below algorithm shows a naive search-control strategy: simply use visited state-action pairs and store them into the search-control queue. During the planning stage, these pairs are uniformly sampled according to the original paper.

The recent works by Pan et al. (2019, 2020) propose two search-control strategies to generate states. The first one is to search high-value states actively, and the second one is to search states whose values are difficult to learn.

However, there are several limitations of the two previous works. First, they do not provide any theoretical justification to use the stochastic gradient ascent trajectories for search-control. Second, HC on gradient norm and Hessian norm of the learned value function (Pan et al., 2020) suffers from great computation cost and zero or explosive gradient due to the high order differentiation (i.e., $\nabla_s||\nabla_s v(s)||$) as suggested by the authors. When using ReLu as activation functions, such high order differentiation almost results in zero gradients. We empirically verified this phenomenon. And this phenomenon can also be verified by intuition from the work by Goodfellow et al. (2015), which suggests that ReLU neural networks are locally almost linear. Then it is not surprising to have zero higher order derivatives. Third, the two methods are prone to result in sub-optimal policies: consider that the values of states are relatively well-learned and fixed, then value-based search-control (Dyna-Value) would still find those high-value states even though they might already have low TD error.

### A.2  DISCUSSION ON THE LANGEVIN DYNAMICS MONTE CARLO METHOD

**Theoretical mechanism**. Define a SDE: $dW(t) = \nabla U(W_t)dt + \sqrt{2}dB_t$, where $B_t \in \mathbb{R}^d$ is a $d$-dimensional Brownian motion and $U$ is a continuous differentiable function. It turns out that the Langevin diffusion $(W_t)_{t\geq 0}$ converges to a unique invariant distribution $p(x) \propto \exp(U(x))$ (Chiang et al., 1987). By applying the Euler-Maruyama discretization scheme to the SDE, we acquire the discretized version $Y_{k+1} = Y_k + \alpha_{k+1}\nabla U(Y_k) + \sqrt{2\alpha_{k+1}}Z_{k+1}$ where $(Z_k)_{k\geq 1}$ is an i.i.d. sequence of standard $d$-dimensional Gaussian random vectors and $(\alpha_k)_{k\geq 1}$ is a sequence of step sizes. It has been proved that the limiting distribution of the sequence $(Y_k)_{k\geq 1}$ converges to the invariant distribution of the underlying SDE (Roberts, 1996; Durmus & Moulines, 2017). As a result, considering $U(\cdot)$ as $\log|\delta(\cdot)|$, $Y$ as $s$ justifies our SGLD sampling method..

---

**Algorithm 2** Tabular Dyna

---

Initialize $Q(s,a)$; initialize model $\mathcal{M}(s,a), \forall (s,a) \in \mathcal{S} \times \mathcal{A}$
**while** true **do**
    observe $s$, take action $a$ by $\epsilon$-greedy w.r.t $Q(s, \cdot)$
    execute $a$, observe reward $R$ and next State $s'$
    Q-learning update for $Q(s,a)$
    update model $\mathcal{M}(s,a)$ (i.e. by counting)
    store $(s,a)$ into search-control queue // this is a naive search-control strategy
    **for** i=1:d **do**
        sample $(\tilde{s}, \tilde{a})$ from search-control queue
        $(\tilde{s}', \tilde{R}) \leftarrow \mathcal{M}(\tilde{s}, \tilde{a})$ // simulated transition
        Q-learning update for $Q(\tilde{s}, \tilde{a})$ // planning updates/steps

---

### A.3 PROOF FOR THEOREM 1

**Theorem 1.** For a constant $c$ determined by $\theta, \mathcal{T}$, we have

$$\mathbb{E}_{(x,y)\sim uniform(\mathcal{T})} \left[ \frac{1}{3} \cdot \frac{\partial |f_\theta(x) - y|^3}{\partial \theta} \right] = c \cdot \mathbb{E}_{(x,y)\sim q(x,y;\theta)} \left[ \frac{1}{2} \cdot \frac{\partial (f_\theta(x) - y)^2}{\partial \theta} \right]$$

*Proof.* For the l.h.s., we have,

$$\mathbb{E}_{(x,y)\sim uniform(\mathcal{T})} \left[ \frac{1}{3} \cdot \frac{\partial |f_\theta(x) - y|^3}{\partial \theta} \right] \tag{4}$$

$$= \frac{1}{3 \cdot n} \cdot \sum_{i=1}^{n} \frac{\partial |f_\theta(x(i)) - y(i)|^3}{\partial \theta} \tag{5}$$

$$= \frac{1}{3 \cdot n} \cdot \sum_{i=1}^{n} \frac{\partial \left( (f_\theta(x(i)) - y(i))^2 \right)^{\frac{3}{2}}}{\partial \theta} \tag{6}$$

$$= \frac{1}{3 \cdot n} \cdot \sum_{i=1}^{n} \frac{\partial \left( (f_\theta(x(i)) - y(i))^2 \right)^{\frac{3}{2}}}{\partial (f_\theta(x) - y)^2} \cdot \frac{\partial (f_\theta(x(i)) - y(i))^2}{\partial \theta} \tag{7}$$

$$= \frac{1}{2 \cdot n} \cdot \sum_{i=1}^{n} |f_\theta(x(i)) - y(i)| \cdot \frac{\partial (f_\theta(x(i)) - y(i))^2}{\partial \theta}. \tag{8}$$

On the other hand, for the r.h.s., we have,

$$\mathbb{E}_{(x,y)\sim q(x,y;\theta)} \left[ \frac{1}{2} \cdot \frac{\partial (f_\theta(x) - y)^2}{\partial \theta} \right] \tag{9}$$

$$= \frac{1}{2} \cdot \sum_{i=1}^{n} q(x_i, y_i; \theta) \cdot \frac{\partial (f_\theta(x(i)) - y(i))^2}{\partial \theta} \tag{10}$$

$$= \frac{n}{\sum_{j=1}^{n} |f_\theta(x_j) - y_j|} \cdot \left[ \frac{1}{2 \cdot n} \cdot \sum_{i=1}^{n} |f_\theta(x(i)) - y(i)| \cdot \frac{\partial (f_\theta(x(i)) - y(i))^2}{\partial \theta} \right] \tag{11}$$

$$= \frac{n}{\sum_{j=1}^{n} |f_\theta(x_j) - y_j|} \cdot \mathbb{E}_{(x,y)\sim uniform(\mathcal{T})} \left[ \frac{1}{3} \cdot \frac{\partial |f_\theta(x) - y|^3}{\partial \theta} \right]. \tag{12}$$

Setting $c = \frac{\sum_{i=1}^{n} |f_\theta(x_i) - y_i|}{n}$ completes the proof. $\qquad\square$

A.4    PROOF FOR THEOREM 2

**Theorem 2.** Let $n$ be a positive integer (i.e., the number of training samples). Let $x_t, \tilde{x}_t \in \mathbb{R}^n$ be the target estimates of all samples at time $t$. Let $x_t(i)(i \in [n], [n] \stackrel{\text{def}}{=} \{1, 2, ..., n\})$ denote the $i$th element in the vector. We define the following notations.

$$\ell_2(x, y) \stackrel{\text{def}}{=} \frac{1}{2} \cdot \sum_{i=1}^{n} (x(i) - y(i))^2, \quad \ell_3(x, y) \stackrel{\text{def}}{=} \frac{1}{3} \cdot \sum_{i=1}^{n} |x(i) - y(i)|^3,$$

$$\delta_t \stackrel{\text{def}}{=} \sum_{i=1}^{n} \delta_t(i) = \sum_{i=1}^{n} |x_t(i) - y(i)|, \quad \tilde{\delta}_t \stackrel{\text{def}}{=} \sum_{i=1}^{n} \tilde{\delta}_t(i) = \sum_{i=1}^{n} |\tilde{x}_t(i) - y(i)|, \ \forall t \geq 0$$

where $y(i) \in \mathbb{R}$ is the training target for the $i$th training sample. Let $\{x_t\}_{t \geq 0}$ and $\{\tilde{x}_t\}_{t \geq 0}$ be generated by using $\ell_2, \ell_3$ objectives respectively. That is, $\forall i \in [n]$,

$$\frac{dx_t(i)}{dt} = -\eta \cdot \frac{d\ell_2(x_t, y)}{dx_t(i)}, \quad \frac{d\tilde{x}_t(i)}{dt} = -\eta \cdot \frac{d\ell_3(\tilde{x}_t, y)}{d\tilde{x}_t(i)}.$$

Assume the same initialization $x_0 = \tilde{x}_0$. Then: **(i)** For all $i \in [n]$, define the following hitting time, which is the minimum time that the absolute error takes to be $\leq \epsilon(i)$,

$$t_\epsilon(i) \stackrel{\text{def}}{=} \min_t \{t \geq 0 : \delta_t(i) \leq \epsilon(i)\}, \quad \tilde{t}_\epsilon(i) \stackrel{\text{def}}{=} \min_t \{t \geq 0 : \tilde{\delta}_t(i) \leq \epsilon(i)\}.$$

Then, $\forall i \in [n]$ s.t. $\delta_0(i) > 1$, given an absolute error threshold $\epsilon(i) \geq 0$, there exists $\epsilon_0(i) \in (0, 1)$, such that for all $\epsilon(i) > \epsilon_0(i), t_\epsilon(i) \geq \tilde{t}_\epsilon(i)$.

**(ii)** Define the following quantity, for all $t \geq 0$,

$$H_t^{-1} \stackrel{\text{def}}{=} \frac{1}{n} \cdot \sum_{i=1}^{n} \frac{1}{\delta_t(i)} = \frac{1}{n} \cdot \sum_{i=1}^{n} \frac{1}{|x_t(i) - y(i)|}. \tag{13}$$

Given any $0 < \epsilon \leq \delta_0 = \sum_{i=1}^{n} \delta_0(i)$, define the following hitting time, which is the minimum time that the total absolute error takes to be $\leq \epsilon$,

$$t_\epsilon \stackrel{\text{def}}{=} \min_t \{t \geq 0 : \delta_t \leq \epsilon\}, \quad \tilde{t}_\epsilon \stackrel{\text{def}}{=} \min_t \{t \geq 0 : \tilde{\delta}_t \leq \epsilon\}. \tag{14}$$

If there exists $\delta_0 \in \mathbb{R}$ and $0 < \epsilon \leq \delta_0$ such that the following holds,

$$H_0^{-1} \leq \frac{\log (\delta_0/\epsilon)}{\frac{\delta_0}{\epsilon} - 1}, \tag{15}$$

then we have, $t_\epsilon \geq \tilde{t}_\epsilon$, which means gradient descent using the cubic loss function will achieve the total absolute error threshold $\epsilon$ faster than using the square loss function.

*Proof.* **First part. (i).** For the $\ell_2$ loss function, for all $i \in [n]$ and $t \geq 0$, we have,

$$\frac{d\delta_t(i)}{dt} = \sum_{j=1}^{n} \frac{d\delta_t(i)}{dx_t(j)} \cdot \frac{dx_t(j)}{dt} \tag{16}$$

$$= \frac{d\delta_t(i)}{dx_t(i)} \cdot \frac{dx_t(i)}{dt} \quad \left(\frac{d\delta_t(i)}{dx_t(j)} = 0 \text{ for all } i \neq j\right) \tag{17}$$

$$= \text{sgn}\{x_t(i) - y(i)\} \cdot (-\eta) \cdot \frac{d\ell_2(x_t, y)}{dx_t(i)} \tag{18}$$

$$= \text{sgn}\{x_t(i) - y(i)\} \cdot (-\eta) \cdot (x_t(i) - y(i)) \tag{19}$$

$$= -\eta |x_t(i) - y(i)| \tag{20}$$

$$= -\eta \cdot \delta_t(i), \tag{21}$$

which implies that,

$$\frac{d\{\log \delta_t(i)\}}{dt} = \frac{1}{\delta_t(i)} \cdot \frac{d\delta_t(i)}{dt} = -\eta. \tag{22}$$

Taking integral, we have,

$$\log \delta_t(i) - \log \delta_0(i) = -\eta \cdot t. \tag{23}$$

Let $\delta_t(i) = \epsilon(i)$. We have,

$$t_\epsilon(i) \stackrel{\text{def}}{=} \frac{1}{\eta} \cdot \log \left( \frac{\delta_0(i)}{\delta_t(i)} \right) = \frac{1}{\eta} \cdot \log \left( \frac{\delta_0(i)}{\epsilon(i)} \right). \tag{24}$$

On the other hand, for the $\ell_3$ loss function, we have,

$$\frac{d\{\tilde{\delta}_t(i)^{-1}\}}{dt} = \sum_{j=1}^n \frac{d\tilde{\delta}_t(i)^{-1}}{d\tilde{x}_t(j)} \cdot \frac{d\tilde{x}_t(j)}{dt} \tag{25}$$

$$= \frac{d\tilde{\delta}_t(i)^{-1}}{d\tilde{x}_t(i)} \cdot \frac{d\tilde{x}_t(i)}{dt} \tag{26}$$

$$= -\frac{1}{\tilde{\delta}_t(i)^2} \cdot \frac{d\tilde{\delta}_t(i)}{d\tilde{x}_t(i)} \cdot \frac{d\tilde{x}_t(i)}{dt} \tag{27}$$

$$= -\frac{1}{(\tilde{x}_t(i) - y(i))^2} \cdot \text{sgn}\{\tilde{x}_t(i) - y(i)\} \cdot (-\eta) \cdot \frac{d\ell_3(\tilde{x}_t, y)}{d\tilde{x}_t(i)} \tag{28}$$

$$= -\frac{\text{sgn}\{\tilde{x}_t(i) - y(i)\}}{(\tilde{x}_t(i) - y(i))^2} \cdot (-\eta) \cdot (\tilde{x}_t(i) - y(i))^2 \cdot \text{sgn}\{\tilde{x}_t(i) - y(i)\} \tag{29}$$

$$= \eta. \tag{30}$$

Taking integral, we have,

$$\frac{1}{\tilde{\delta}_t(i)} - \frac{1}{\tilde{\delta}_0(i)} = \eta \cdot t. \tag{31}$$

Let $\tilde{\delta}_t(i) = \epsilon(i)$. We have,

$$\tilde{t}_\epsilon(i) \stackrel{\text{def}}{=} \frac{1}{\eta} \cdot \left( \frac{1}{\tilde{\delta}_t(i)} - \frac{1}{\tilde{\delta}_0(i)} \right) = \frac{1}{\eta} \cdot \left( \frac{1}{\epsilon(i)} - \frac{1}{\tilde{\delta}_0(i)} \right). \tag{32}$$

Then we have,

$$t_\epsilon(i) - \tilde{t}_\epsilon(i) = \frac{1}{\eta} \cdot \log \left( \frac{\delta_0(i)}{\epsilon(i)} \right) - \frac{1}{\eta} \cdot \left( \frac{1}{\epsilon(i)} - \frac{1}{\tilde{\delta}_0(i)} \right) \tag{33}$$

$$= \frac{1}{\eta} \cdot \left[ \left( \log \frac{1}{\epsilon(i)} - \frac{1}{\epsilon(i)} \right) - \left( \log \frac{1}{\delta_0(i)} - \frac{1}{\tilde{\delta}_0(i)} \right) \right]. \tag{34}$$

According to $x_0(i) = \tilde{x}_0(i)$, we have

$$\delta_0(i) = |x_t(i) - y(i)| \tag{35}$$

$$= |\tilde{x}_t(i) - y(i)| \tag{36}$$

$$= \tilde{\delta}_0(i). \tag{37}$$

Define the following function, for all $x > 0$,

$$f(x) = \log \frac{1}{x} - \frac{1}{x}. \tag{38}$$

We have, the continuous function $f$ is monotonically increasing for $x \in (0, 1]$ and monotonically decreasing for $x \in (1, \infty)$. Also, note that, $\max_{x>0} f(x) = f(1) = -1$, $\lim_{x \to 0} f(x) = \lim_{x \to \infty} f(x) = -\infty$.

Given $\delta_0(i) = \tilde{\delta}_0(i) > 1$, we have $f(\delta_0(i)) < f(1) = -1$. According to the intermediate value theorem, there exists $\epsilon_0(i) \in (0, 1)$, such that $f(\epsilon_0(i)) = f(\delta_0(i))$. Since $f(\cdot)$ is monotonically

increasing on $(0, 1]$ and monotonically decreasing on $(1, \infty)$, for all $\epsilon(i) \in [\epsilon_0(i), \delta_0(i)]$, we have $f(\epsilon(i)) \geq f(\delta_0(i))$[2]. Therefore, we have,

$$t_\epsilon(i) - \tilde{t}_\epsilon(i) = \frac{1}{\eta} \cdot (f(\epsilon(i)) - f(\delta_0(i))) \geq 0. \tag{39}$$

**Second part. (ii).** For the square loss function, we have, for all $t \geq 0$,

$$\frac{d\delta_t}{dt} = \sum_{i=1}^{n} \frac{d\delta_t(i)}{dt} \tag{40}$$

$$= -\eta \cdot \sum_{i=1}^{n} \delta_t(i) \qquad \text{(by eq. (16))} \tag{41}$$

$$= -\eta \cdot \delta_t, \tag{42}$$

which implies that,

$$\frac{d\{\log \delta_t\}}{dt} = \frac{1}{\delta_t} \cdot \frac{d\delta_t}{dt} = -\eta. \tag{43}$$

Taking integral, we have,

$$\log \delta_t - \log \delta_0 = -\eta \cdot t. \tag{44}$$

Let $\delta_t = \epsilon$. We have,

$$t_\epsilon \overset{\text{def}}{=} \frac{1}{\eta} \cdot \log\left(\frac{\delta_0}{\delta_t}\right) = \frac{1}{\eta} \cdot \log\left(\frac{\delta_0}{\epsilon}\right). \tag{45}$$

After $t_\epsilon$ time, for all $i \in [n]$, we have,

$$\delta_{t_\epsilon}(i) = \delta_0(i) \cdot \exp\{-\eta \cdot t_\epsilon\}. \tag{46}$$

On the other hand, for the cubic loss function, we have, for all $t \geq 0$,

$$\frac{dH_t^{-1}}{dt} = \frac{1}{n} \cdot \sum_{i=1}^{n} \frac{d\{\tilde{\delta}_t(i)^{-1}\}}{dt} \tag{47}$$

$$= \eta. \qquad \text{(by eq. (25))} \tag{48}$$

Taking integral, we have,

$$H_t^{-1} - H_0^{-1} = \eta \cdot t, \tag{49}$$

which means given a $H_t^{-1}$ value, we can calculate the hitting time as,

$$t = \frac{1}{\eta} \cdot \left(H_t^{-1} - H_0^{-1}\right). \tag{50}$$

Now consider after $t_\epsilon$ time, using gradient descent with the square loss function we have $\delta_{t_\epsilon}(i) = \delta_0(i) \cdot \exp\{-\eta \cdot t_\epsilon\}$ for all $i \in [n]$, which corresponds to,

$$H_{t_\epsilon}^{-1} = \frac{1}{n} \cdot \sum_{i=1}^{n} \frac{1}{\delta_{t_\epsilon}(i)} \tag{51}$$

$$= \frac{1}{n} \cdot \sum_{i=1}^{n} \frac{1}{\delta_0(i) \cdot \exp\{-\eta \cdot t_\epsilon\}}. \qquad \text{(by eq. (46))} \tag{52}$$

---

[2]Note that $\epsilon(i) < \delta_0(i)$ by the design of using gradient descent updating rule. If the two are equal, $t_\epsilon(i) = \tilde{t}_\epsilon(i) = 0$ holds trivially.

Therefore, the hitting time of using gradient descent with the cubic loss function to achieve the $H_{t_\epsilon}^{-1}$ value is,

$$\tilde{t}_\epsilon = \frac{1}{\eta} \cdot \left( H_{t_\epsilon}^{-1} - H_0^{-1} \right) \tag{53}$$

$$= \frac{1}{\eta} \cdot \left( \frac{1}{n} \cdot \sum_{i=1}^{n} \frac{1}{\delta_0(i) \cdot \exp\{-\eta \cdot t_\epsilon\}} - \frac{1}{n} \cdot \sum_{i=1}^{n} \frac{1}{\delta_0(i)} \right) \tag{54}$$

$$= \frac{1}{\eta} \cdot (\exp\{\eta \cdot t_\epsilon\} - 1) \cdot H_0^{-1} \tag{55}$$

$$\leq \frac{1}{\eta} \cdot (\exp\{\eta \cdot t_\epsilon\} - 1) \cdot \frac{\log(\delta_0/\epsilon)}{\frac{\delta_0}{\epsilon} - 1} \qquad \text{(by eq. (15))} \tag{56}$$

$$= \frac{1}{\eta} \cdot \left( \frac{\delta_0}{\epsilon} - 1 \right) \cdot \frac{\log(\delta_0/\epsilon)}{\frac{\delta_0}{\epsilon} - 1} \tag{57}$$

$$= \frac{1}{\eta} \cdot \log\left( \frac{\delta_0}{\epsilon} \right) \tag{58}$$

$$= t_\epsilon, \qquad \text{(by eq. (45))} \tag{59}$$

finishing the proof. $\square$

**Remark.** Figure 7 shows the function $f(x) = \ln\frac{1}{x} - \frac{1}{x}, x > 0$. Fix arbitrary $x' > 1$, there will be another root $\epsilon_0 < 1$ s.t. $f(\epsilon_0) = f(x')$. However, there is no real-valued solution for $\epsilon_0$. The solution in $\mathbb{C}$ is $\epsilon_0 = -\frac{1}{W(\log 1/\delta_0 - 1/\delta_0 - \pi i)}$, where $W(\cdot)$ is a Wright Omega function. Hence, finding the exact value of $\epsilon_0$ would require a definition of ordering on complex plane. Our current theorem statement is sufficient for the purpose of characterizing convergence rate. The theorem states that there always exists some desired low error level $< 1$, minimizing the square loss converges slower than the cubic loss.

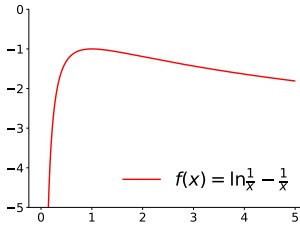

Figure 7: The function $f(x) = \ln\frac{1}{x} - \frac{1}{x}, x > 0$. The function reaches maximum at $x = 1$.

**Simulations.** The theorem says that if we want to minimize our loss function to certain small nonzero error level, the cubic loss function offers faster convergence rate. Intuitively, cubic loss provides sharper gradient information when the loss is large as shown in Figure 8(a)(b). Here we provides a simulation. Consider the following minimization problems: $\min_{x\geq 0} x^2$ and $\min_{x\geq 0} x^3$. We use the hitting time formulae $t_\epsilon = \frac{1}{\eta} \cdot \ln\left\{ \frac{\delta_0}{\epsilon} \right\}, \tilde{t}_\epsilon = \frac{1}{\eta} \cdot \left( \frac{1}{\epsilon} - \frac{1}{\delta_0} \right)$ derived in the proof, to compute the hitting time ratio $\frac{t_\epsilon}{\tilde{t}_\epsilon}$ under different initial values $x_0$ and final error value $\epsilon$. In Figure 8(c)(d), we can see that it usually takes a significantly shorter time for the cubic loss to reach a certain $x_t$ with various initial $x_0$ values.

## A.5 Error Bound between Sampling Distributions

We now provide the error bound between the sampling distribution estimated by using a true model and a learned model. We denote the transition probability distribution under policy $\pi$ and the true model as $\mathcal{P}^\pi(r, s'|s)$, and the learned model as $\hat{\mathcal{P}}^\pi(r, s'|s)$. Let $p(s)$ and $\hat{p}(s)$ be the convergent distributions described in the above sampling method by using the true and learned models respectively. Let $d_{tv}(\cdot, \cdot)$ be the total variation distance between the two probability distributions.

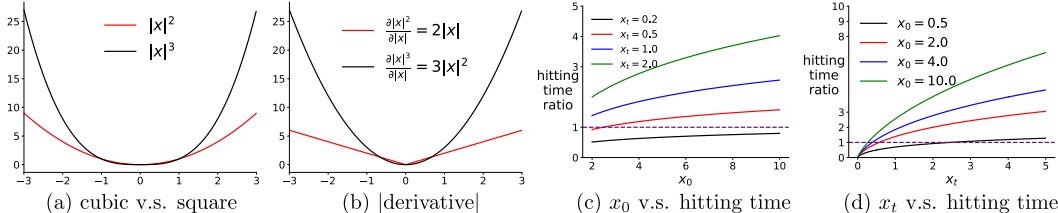

Figure 8: (a) show cubic v.s. square function. (b) shows their absolute derivatives. (c) shows the hitting time ratio v.s. initial value $x_0$ under different target value $x_t$. (d) shows the ratio v.s. the target $x_t$ to reach under different $x_0$. Note that a ratio larger than 1 indicates a longer time to reach the given $x_t$ for the square loss.

Define $u(s) \stackrel{\text{def}}{=} |\delta(s, y(s))|$, $\hat{u}(s) \stackrel{\text{def}}{=} |\delta(s, \hat{y}(s))|$, $Z \stackrel{\text{def}}{=} \int_{s \in \mathcal{S}} u(s)ds$, $\hat{Z} \stackrel{\text{def}}{=} \int_{s \in \mathcal{S}} \hat{u}(s)ds$. Then we have the following bound.

**Theorem 3.** *Assume: 1) the reward magnitude is bounded* $|r| \leq R_{max}$ *and define* $V_{max} \stackrel{\text{def}}{=} \frac{R_{max}}{1-\gamma}$; *2) the largest model error for a single state is* $\epsilon_s \stackrel{\text{def}}{=} \max_s d_{tv}(\mathcal{P}^\pi(\cdot|s), \hat{\mathcal{P}}^\pi(\cdot|s))$ *and the to- tal model error is bounded, i.e.* $\epsilon \stackrel{\text{def}}{=} \int_{s \in \mathcal{S}} \epsilon_s ds < \infty$. *Then* $\forall s \in \mathcal{S}, |p(s) - \hat{p}(s)| \leq \min(\frac{V_{max}(p(s)\epsilon + \epsilon_s)}{\hat{Z}}, \frac{V_{max}(\hat{p}(s)\epsilon + \epsilon_s)}{Z})$.

*Proof.* First, we bound the estimated temporal difference error. Fix an arbitrary state $s \in \mathcal{S}$, it is sufficient the consider the case $u(s) > \hat{u}(s)$, then

$$|u(s) - \hat{u}(s)| = u(s) - \hat{u}(s)$$
$$= \mathbb{E}_{(r,s') \sim \mathcal{P}^\pi}[r + \gamma v^\pi(s')] - \mathbb{E}_{(r,s') \sim \hat{\mathcal{P}}^\pi}[r + \gamma v^\pi(s')]$$
$$= \int_{s,r} (r + \gamma v^\pi(s))(\mathcal{P}^\pi(s', r|s) - \hat{\mathcal{P}}^\pi(s', r|s))ds'dr$$
$$\leq (R_{max} + \gamma \frac{R_{max}}{1-\gamma}) \int_{s,r} (\mathcal{P}^\pi(s', r|s) - \hat{\mathcal{P}}^\pi(s', r|s))ds'dr$$
$$\leq V_{max} d_{tv}(\mathcal{P}^\pi(\cdot|s), \hat{\mathcal{P}}^\pi(\cdot|s)) \leq V_{max} \epsilon_s$$

Now, we show that $|Z - \hat{Z}| \leq V_{max}\epsilon$.

$$|Z - \hat{Z}| = |\int_{s \in \mathcal{S}} u(s)ds - \int_{s \in \mathcal{S}} \hat{u}(s)ds| = |\int_{s \in \mathcal{S}} (u(s) - \hat{u}(s))ds|$$
$$\leq \int_{s \in \mathcal{S}} |u(s) - \hat{u}(s)|ds \leq V_{max} \int_{s \in \mathcal{S}} \epsilon_s ds = V_{max}\epsilon$$

Consider the case $p(s) > \hat{p}(s)$ first.

$$p(s) - \hat{p}(s) = \frac{u(s)}{Z} - \frac{\hat{u}(s)}{\hat{Z}}$$
$$\leq \frac{u(s)}{Z} - \frac{u(s) - V_{max}\epsilon_s}{\hat{Z}} = \frac{u(s)\hat{Z} - u(s)Z + ZV_{max}\epsilon_s}{Z\hat{Z}}$$
$$\leq \frac{u(s)V_{max}\epsilon + ZV_{max}\epsilon_s}{Z\hat{Z}} = \frac{V_{max}(p(s)\epsilon + \epsilon_s)}{\hat{Z}}$$

Meanwhile, below inequality should also hold:

$$p(s) - \hat{p}(s) = \frac{u(s)}{Z} - \frac{\hat{u}(s)}{\hat{Z}} \leq \frac{\hat{u}(s) + V_{max}\epsilon_s}{Z} - \frac{\hat{u}(s)}{\hat{Z}}$$
$$= \frac{\hat{u}(s)\hat{Z} - \hat{u}(s)Z + \hat{Z}V_{max}\epsilon_s}{Z\hat{Z}} \leq \frac{V_{max}(\hat{p}(s)\epsilon + \epsilon_s)}{Z}$$

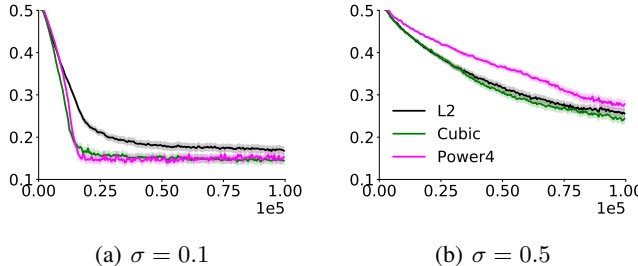

(a) $\sigma = 0.1$        (b) $\sigma = 0.5$

Figure 9: Figure(a)(b) show the testing RMSE as a function of number of mini-batch updates with increasing noise standard deviation $\sigma$ added to the training targets. We compare the performances of **Power4(magenta)**, **L2 (black)**, **Cubic (forest green)**. The results are averaged over 50 random seeds. The shade indicates standard error. Note that the testing set is not noise-contaminated.

Because both the two inequalities must hold, when $p(s) - \hat{p}(s) > 0$, we have:

$$p(s) - \hat{p}(s) \leq \min(\frac{V_{max}(p(s)\epsilon + \epsilon_s)}{\hat{Z}}, \frac{V_{max}(\hat{p}(s)\epsilon + \epsilon_s)}{Z})$$

It turns out that the bound is the same when $p(s) \leq \hat{p}(s)$. This completes the proof. $\qquad\square$

### A.6 HIGH POWER LOSS FUNCTIONS

We would like to point out that directly using a high power objective in general problems is unlikely to have an advantage.

First, notice that our convergence rate is characterized w.r.t. to the expected updating rule, not stochastic gradient updating rule. When using a stochastic sample to estimate the gradient, high power objectives are sensitive to the outliers as they augment the effect of noise. Robustness to outliers is also the motivation behind the Huber loss (Huber, 1964) which, in fact, uses low power error in most places so it can be less sensitive to outliers.

We conduct experiments to examine the effect of noise on using high power objectives. We use the same dataset as described in Section 3.3. We use a training set with 4k training examples. The naming rules are as follows. **Cubic** is minimizing the cubic objective (i.e. $\min_\theta \frac{1}{n} \sum_{i=1}^{n} |f_\theta(x_i) - y_i|^3$) by uniformly sampling, and **Power4** is $\min_\theta \frac{1}{n} \sum_{i=1}^{n} (f_\theta(x_i) - y_i)^4$ by uniformly sampling.

Figure 9 (a)(b) shows the learning curves of uniformly sampling for Cubic and for Power4 trained by adding noises with standard deviation $\sigma = 0.1, 0.5$ respectively to the training targets. It is not surprising that all algorithms learn slower when we increase the noise variance added to the target variables. However, one can see that *high power objectives is more sensitive to noise variance added to the targets than the regular L2*: when $\sigma = 0.1$, the higher power objectives perform better than the regular L2; after increasing $\sigma$ to $0.5$, Cubic becomes almost the same as L2, while Power4 becomes worse than L2.

Second, it should be noted that in our theorem, we do not characterize the convergence rate to the minimum; instead, we show the convergence rate to a certain low error solution, corresponding to early learning performance. In optimization literature, it is known that cubic power would converge slower to the minimizer as it has a relatively flat bottom. However, it may be an interesting future direction to study how to combine objectives with different powers so that optimizing the hybrid objective leads to a faster convergence rate to the optimum and is robust to outliers.

### A.7 ADDITIONAL EXPERIMENTS

In this section, we include the following additional experimental results:

1. As a supplementary to Figure 1 from Section 3.3, we show the learning performance measured by training errors to show the negative effects of the two limitations.

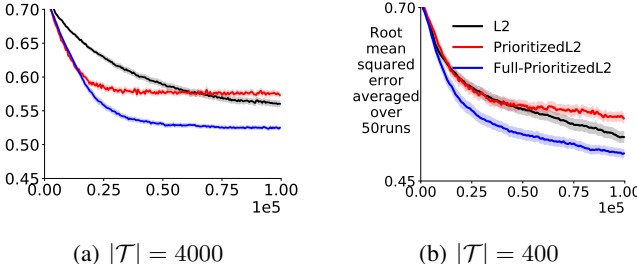

(a) $|\mathcal{T}| = 4000$          (b) $|\mathcal{T}| = 400$

Figure 10: Figure (a)(b) show the training RMSE as a function of number of mini-batch updates with a training set containing $4k$ examples and another containing $400$ examples respectively. We compare the performances of **Full-PrioritizedL2 (blue)**, **L2 (black)**, and **PrioritizedL2 (red)**. The results are averaged over $50$ random seeds. The shade indicates standard error.

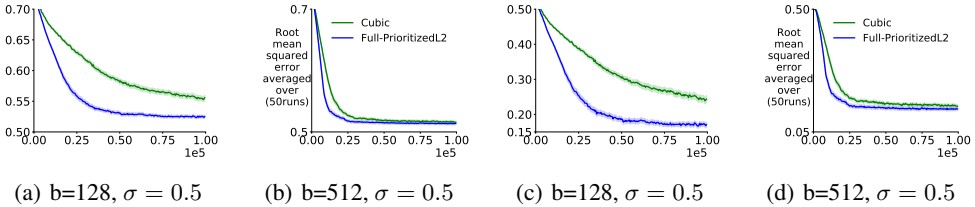

(a) b=128, $\sigma = 0.5$    (b) b=512, $\sigma = 0.5$    (c) b=128, $\sigma = 0.5$    (d) b=512, $\sigma = 0.5$

Figure 11: Figure(a)(b) show the training RMSE as a function of number of mini-batch updates with increasing mini-batch size $b$. Figure (c)(d) show the testing RMSE. We compare the performances of **Full-PrioritizedL2 (blue)**, **Cubic (forest green)**. As we increase the mini-batch size, the two performs more similar to each other. The results are averaged over $50$ random seeds. The shade indicates standard error.

2. Empirical verification of Theorem 1 (prioritized sampling and uniform sampling on cubic power equivalence).

3. Results on MazeGridWorld from Pan et al. (2020).

### A.7.1 TRAINING ERROR CORRESPONDING TO FIGURE 1 FROM SECTION 3.3

Note that our Theorem 1 and 2 characterize the expected gradient calculated on the training set; hence it is sufficient to examine the learning performances measured by training errors. However, the testing error is usually the primary concern, so we put the testing error in the main body. As a sanity check, we also investigate the learning performances measured by training error and find that those algorithms behave similarly as shown in Figure 10 where the algorithms are trained by using training sets with decreasing training examples from (a) to (b). As we reduce the training set size, Full-PrioritizedL2 is closer to L2. Furthermore, PrioritizedL2 is always worse than Full-PrioritizedL2. These observations show the negative effects resulting from the issues of outdated priorities and insufficient sample space coverage.

### A.7.2 EMPIRICAL VERIFICATION OF THEOREM 1

Theorem 1 states that the expected gradient of doing prioritized sampling on mean squared error is equal to the gradient of doing uniformly sampling on cubic power loss. As a result, we expect that the learning performance on the training set (note that we calculate gradient by using training examples) should be similar when we use a large mini-batch update as the estimate of the expectation terms become close.

We use the same dataset as described in Section 3.3 and keep using training size $4k$. Figure 11(a)(b) shows that when we increase the mini-batch size, *the two algorithms Full-PrioritizedL2 and Cubic are becoming very close to each other*, verifying our theorem.

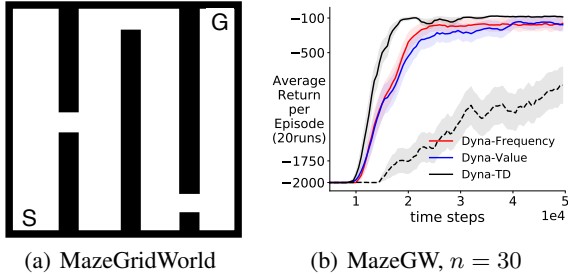

(a) MazeGridWorld       (b) MazeGW, $n = 30$

Figure 12: Figure(a) shows MazeGridWorld(GW) taken from Pan et al. (2020) and the learning curves are in (b). We show evaluation learning curves of **Dyna-TD (black)**, **Dyna-Frequency (red)**, and **Dyna-Value (blue)**. The dashed line indicates Dyna-TD trained with an online learned model. All results are averaged over 20 random seeds after smoothing over a window of size 30. The shade indicates standard error.

Note that our theorem characterizes the expected gradient calculated on the training set; hence it is sufficient to examine the learning performances measured by training errors. However, usually, the testing error is the primary concern. For completeness, we also investigate the learning performances measured by testing error and find that the tested algorithms behave similarly as shown in Figure 11(c)(d).

### A.7.3 RESULTS ON MAZEGRIDWORLD DOMAIN

In Figure 12, we demonstrate that our algorithm can work better than Dyna-Frequency on a Maze-GridWorld domain (Pan et al., 2020), where Dyna-Frequency was shown to be superior to Dyna-Value and model-free baselines. This result further confirms the usefulness of our sampling approach.

### A.8 REPRODUCIBLE RESEARCH

Our implementations are based on tensorflow with version 1.13.0 (Abadi et al., 2015). We use Adam optimizer (Kingma & Ba, 2014) for all experiments.

### A.8.1 REPRODUCE EXPERIMENTS BEFORE SECTION 5

**Supervised learning experiment.** For the supervised learning experiment shown in section 3, we use $32 \times 32$ tanh units neural network, with learning rate swept from $\{0.01, 0.001, 0.0001, 0.00001\}$ for all algorithms. We compute the constant $c$ as specified in the Theorem 1 at each time step for Cubic loss. We compute the testing error every 500 iterations/mini-batch updates and our evaluation learning curves are plotted by averaging 50 random seeds. For each random seed, we randomly split the dataset to testing set and training set and the testing set has 1k data points. Note that the testing set is not noise-contaminated.

**Reinforcement Learning experiments in Section 3.** We use a particularly small neural network $16 \times 16$ to highlight the issue of incomplete priority updating. Intuitively, a large neural network may be able to memorize each state's value and thus updating one state's value is less likely to affect others. We choose a small neural network, in which case a complete priority updating for all states should be very important. We set the maximum ER buffer size as 10k and mini-batch size as 32. The learning rate is chosen from $\{0.0001, 0.001\}$ and the target network is updated every 1k steps.

**Distribution distance computation in Section 4.** We now introduce the implementation details for Figure 3. The distance is estimated by the following steps. First, in order to compute the desired sampling distribution, we discretize the domain into $50 \times 50$ grids and calculate the absolute TD error of each grid (represented by the left bottom vertex coordinates) by using the true environment model and the current learned $Q$ function. We then normalize these priorities to get probability distribution $p^*$. Note that this distribution is considered as the desired one since we have access to all states across the state space with priorities computed by current Q-function at each time step.

Second, we estimate our sampling distribution by randomly sampling 3k states from search-control queue and count the number of states falling into each discretized grid and normalize these counts to get $p_1$. Third, for comparison, we estimate the sampling distribution of the conventional prioritized ER (Schaul et al., 2016) by sampling 3k states from the prioritized ER buffer and count the states falling into each grid and compute its corresponding distribution $p_2$ by normalizing the counts. Then we compute the distances of $p_1, p_2$ to $p^*$ by two weighting schemes: 1) on-policy weighting: $\sum_{j=1}^{2500} d^\pi(s_j)|p_i(s_j) - p^*(s_j)|, i \in \{1, 2\}$, where $d^\pi$ is approximated by uniformly sample 3k states from a recency buffer and normalizing their visitation counts on the discretized GridWorld; 2) uniform weighting: $\frac{1}{2500} \sum_{j=1}^{2500} |p_i(s_j) - p^*(s_j)|, i \in \{1, 2\}$. We examine the two weighting schemes because of two considerations: for the on-policy weighting, we concern about the asymptotic convergent behavior and want to down-weight those states with relatively high TD error but get rarely visited as the policy gets close to optimal; uniform weighting makes more sense during early learning stage, where we consider all states are equally important and want the agents to sufficiently explore the whole state space.

**Computational cost v.s. performance in Section 4.** The setting is the same as we used for Section 5. We use plan step/updates=10 to generate that learning curve.

### A.8.2  REPRODUCE EXPERIMENTS IN SECTION 5

For our algorithm, the pseudo-code with concrete parameter settings is presented in Algorithm 4.

**Common settings.** For all discrete control domains other than roundabout-v0, we use $32 \times 32$ neural network with ReLu hidden units except the Dyna-Frequency which uses tanh units as suggested by the author (Pan et al., 2020). This is one of its disadvantages: the search-control of Dyna-Frequency requires the computation of Hessian-gradient product and it is empirically observed that the Hessian is frequently zero when using ReLu as hidden units. Except the output layer parameters which were initialized from a uniform distribution $[-0.003, 0.003]$, all other parameters are initialized using Xavier initialization (Glorot & Bengio, 2010). We use mini-batch size $b = 32$ and maximum ER buffer size 50k. All algorithms use target network moving frequency 1000 and we sweep learning rate from $\{0.001, 0.0001\}$. We use warm up steps $= 5000$ (i.e. random action is taken in the first 5k time steps) to populate the ER buffer before learning starts. We keep exploration noise as 0.1 without decaying.

**Hyper-parameter settings.** Across RL experiments including both discrete and continuous control tasks, we are able to fix the same parameters for our hill climbing updating rule 3 $s \leftarrow s + \alpha_h \nabla_s \log |\hat{y}(s) - \max_a Q(s, a; \theta_t)| + X$, where we fix $\alpha_h = 0.1, X \sim N(0, 0.01)$.

For our algorithm Dyna-TD, we are able to keep the same parameter setting across all discrete domains: $c = 20$ and learning rate 0.001. For all Dyna variants, we fetch the same number of states ($m = 20$) from hill climbing (i.e. search-control process) as Dyna-TD does, and use $\epsilon_{accept} = 0.1$ and set the maximum number of gradient step as $k = 100$ unless otherwise specified.

Our Prioritized ER is implemented as the proportional version with sum tree data structure. To ensure fair comparison, since all model-based methods are using mixed mini-batch of samples, we use prioritized ER without importance ratio but half of mini-batch samples are uniformly sampled from the ER buffer as a strategy for bias correction. For Dyna-Value and Dyna-Frequency, we use the setting as described by the original papers.

For the purpose of learning an environment model on those discrete control domains, we use a $64 \times 64$ ReLu units neural network to predict $s' - s$ and reward given a state-action pair $s, a$; and we use mini-batch size 128 and learning rate 0.0001 to minimize the mean squared error objective for training the environment model.

**Environment-specific settings.** All of the environments are from OpenAI (Brockman et al., 2016) except that: 1) the GridWorld envirnoment is taken from Pan et al. (2019) and the MazeGridWorld is from Pan et al. (2020); 2) Roundabout-v0 is from (Leurent et al., 2019). For all OpenAI environments, we use the default setting except on Mountain Car where we set the episodic length limit to 2k. The GridWorld has state space $\mathcal{S} = [0, 1]^2$ and each episode starts from the left bottom and the goal area is at the top right $[0.95, 1.0]^2$. There is a wall in the middle with a hole to allow the agent to pass. MazeGridWorld is a more complicated version where the state and action spaces are the same as

---

**Algorithm 3** Dyna-TD

---

**Input:** $m$: number of states to fetch through search-control; $B_{sc}$: empty search-control queue; $B_{er}$: ER buffer; $\epsilon_{accept}$: threshold for accepting a state; initialize $Q$-network $Q_\theta$

**for** $t = 1, 2, \ldots$ **do**

    Observe $(s_t, a_t, s_{t+1}, r_{t+1})$ and add it to $B_{er}$

    // Hill climbing on absolute TD error

    Sample $s$ from $B_{er}$, $c \leftarrow 0$, $\tilde{s} \leftarrow s$

    **while** $c < m$ **do**

        $\hat{y} \leftarrow \mathbb{E}_{s', r \sim \hat{\mathcal{P}}(\cdot|s,a)}[r + \gamma \max_a Q_\theta(s', a)]$

        Update $s$ by rule (3)

        **if** $s$ is out of the state space **then**

            Sample $s$ from $B_{er}$, $\tilde{s} \leftarrow s$ // restart

            **continue**

        **if** $\|\tilde{s} - s\|_2/\sqrt{d} \geq \epsilon_{accept}$ **then**

            // $d$ is the number of state variables, i.e. $\mathcal{S} \subset \mathbb{R}^d$

            Add $s$ into $B_{sc}$, $\tilde{s} \leftarrow s$, $c \leftarrow c + 1$

    //$n$ planning updates

    **for** $n$ times **do**

        Sample a mixed mini-batch with half samples from $B_{sc}$ and half from $B_{er}$

        Update $Q$-network parameters by using the mixed mini-batch

---

GridWorld, but there are two walls in the middle and it takes a long time for model-free methods to be successful. On the this domain, we use the same setting as the original paper for all Dyna variants. We use exactly the same setting as described above except that we change the $Q-$ network size to $64 \times 64$ ReLu units, and number of search-control samples is $m = 50$ as used by the original paper. We refer readers to the original paper (Pan et al., 2020) for more details.

On roundabout-v0 domain, we use $64 \times 64$ ReLu units for all algorithms and set mini-batch size as $64$. The environment model is learned by using a $200 \times 200$ ReLu neural network trained by the same way mentioned above. For Dyna-TD, we start using the model after 5k steps and set $m = 100, k = 500$ and we do search-control every 50 environment time steps to reduce computational cost. To alleviate the effect of model error, we use only 16 out of 64 samples from the search-control queue in a mini-batch.

On Mujoco domains Hopper and Walker2d, we use $200 \times 100$ ReLu units for all algorithms and set mini-batch size as $64$. The environment model is learned by using a $200 \times 200$ ReLu neural network trained by the same way mentioned above. For Dyna-TD, we start using the model after 10k steps and set $m = 100, k = 500$ and we do search-control every 50 environment time steps to reduce computational cost. To alleviate the effect of model error, we use only 16 out of 64 samples from the search-control queue in a mini-batch.

---

**Algorithm 4** Dyna-TD with implementation details

---

**Input or notations:** $k = 20$: number search-control states to acquire by hill climbing, $k_b = 100$: the budget of maximum number of hill climbing steps; $\rho = 0.5$: percentage of samples from search-control queue, $d : \mathcal{S} \subset \mathbb{R}^d$; empty search-control queue $B_{sc}$ and ER buffer $B_{er}$

empirical covariance matrix: $\hat{\Sigma}_s \leftarrow \mathbf{I}$

$\mu_{ss} \leftarrow \mathbf{0} \in \mathbb{R}^{d \times d}, \mu_s \leftarrow \mathbf{0} \in \mathbb{R}^d$ (auxiliary variables for computing empirical covariance matrix, sample average will be maintained for $\mu_{ss}, \mu_s$)

$n_\tau \leftarrow 0$: count for parameter updating times, $\tau \leftarrow 1000$ target network updating frequency

$\epsilon_{accept} \leftarrow 0$: threshold for accepting a state

Initialize $Q$ network $Q_\theta$ and target $Q$ network $Q_{\theta'}$

**for** $t = 1, 2, \ldots$ **do**

    Observe $(s, a, s', r)$ and add it to $B_{er}$

    $\mu_{ss} \leftarrow \frac{\mu_{ss}(t-1) + ss^\top}{t}, \mu_s \leftarrow \frac{\mu_s(t-1) + s}{t}$

    $\hat{\Sigma}_s \leftarrow \mu_{ss} - \mu_s \mu_s^\top$

    $\epsilon_{accept} \leftarrow (1 - \beta)\epsilon_{accept} + \beta ||s' - s||_2$ for $\beta = 0.001$

    // Hill climbing on absolute TD error

    Sample $s$ from $B_{er}, c \leftarrow 0, \tilde{s} \leftarrow s, i \leftarrow 0$

    **while** $c < k$ and $i < k_b$ **do**

        // since environment is deterministic, the environment model becomes a Dirac-delta distribution and we denote it as a deterministic function $\mathcal{M} : \mathcal{S} \times \mathcal{A} \mapsto \mathcal{S} \times \mathbb{R}$

        $s', r \leftarrow \mathcal{M}(s, a)$

        $\hat{y} \leftarrow r + \gamma \max_a Q_\theta(s', a)$

        // add a smooth constant $10^{-5}$ inside the logarithm

        $s \leftarrow s + \alpha_h \nabla_s \log(|\hat{y} - \max_a Q(s, a; \theta_t)| + 10^{-5}) + X, X \sim N(0, 0.01\hat{\Sigma}_s)$

        **if** $s$ is out of the state space **then**

            // restart hill climbing

            Sample $s$ from $B_{er}, \tilde{s} \leftarrow s$

            **continue**

        **if** $||\tilde{s} - s||_2 / \sqrt{d} \geq \epsilon_{accept}$ **then**

            Add $s$ into $B_{sc}, \tilde{s} \leftarrow s, c \leftarrow c + 1$

        $i \leftarrow i + 1$

    **for** $n$ times **do**

        Sample a mixed mini-batch $b$, with proportion $\rho$ from $B_{sc}$ and $1 - \rho$ from $B_{er}$

        Update parameters $\theta$ (i.e. DQN update) with $b$

        $n_\tau \leftarrow n_\tau + 1$

        **if** $mod(n_\tau, \tau) == 0$ **then**

            $Q_{\theta'} \leftarrow Q_\theta$

---

