# OpenReview forum: "Beyond Prioritized Replay: Sampling States in Model-Based Reinforcement Learning via Simulated Priorities"
_ICLR.cc/2022/Conference — ICLR 2022 Submitted_

### Official Review · Reviewer_VdDV · 2021-11-01

**Correctness:** 3
**Technical Novelty And Significance:** 2
**Empirical Novelty And Significance:** 2
**Recommendation:** 3
**Confidence:** 5

**Main Review:**

## Claims, contributions, and significance

There are actually two claims in the paper.
The first (section 3), which I judge as minor and clumsily presented, is that sampling according to the TD error is justified by the fact that it is equivalent to optimizing a cubic loss.
The second is IMHO the main contribution, consists in selecting states for minibatches according to a Langevin dynamic, then exploiting a model to build target values, and finally performing SGD using these samples. This claim could be made clearer since the beginning of the paper (currently, what the algorithm actually is is only clarified in section 4, on page 6).
This second claim takes the form of an algorithm, building on previous work (HC-Dyna) and could be a significant contribution to the community if it were better discussed.

## Clarity

I believe the paper could be improved a lot by verifying the phrasing, the typos and the syntax (all across the paper and mostly in sections 4 and 5).
Some words are regularly missing in sentences. The paper remains readable but is clearly not polished.
Although the claims appear rather clearly in the introduction, the discussion quickly becomes clumsy and difficult to follow. Some notations are also confusing (the ones in thm 2 and in section 4.1 for instance) and sometimes two different notations are used for the same notion across the paper (targets in section 3 are not the same as in section 4, $v(s)$ in 4.1 is actually $v(s,\theta_t)$, etc.).
Also I fail to understand why section 4.2 is not part of section 5.

## Proper grounding would make section 3 a lot better

I would like to encourage the authors to ground their paper in the proper framework they are working in. What is at stake here is the resolution of the Bellman equation via approximate dynamic programming, where the approximation is solved by stochastic gradient descent. No mention of ADP is made whatsoever in the paper. Despite it being the key background of what is done here.
To the authors: Don't get me wrong, this is not a whim, I strongly believe taking the time to properly ground your discussion will help you substantially improve both your contribution and your paper.
For instance, ADP's error bound for the L2 loss have been extensively studied in (Munos 2003, 2005, Scherrer et al 2012) and there is a whole literature on MDP solving (model-based RL is just the current fancy name).
Anecdotally, in the introduction stating that "in particular, prioritized sweeping improves upon vanilla Dyna" induces confusion. PS is a pure dynamic programming scheme (an asynchronous DP one to be precise), it has nothing to do with SGD in ADP. And since it uses tabular representations, it does not need perform SGD steps at all. It loosely served as an inspiration for PER, that's all. It is thus debatable to compare it with Dyna (which is an architecture, rather than an algorithm).

Not setting up properly the ADP loss makes the contribution very brittle and shallow. What is the loss optimized?
The only place we come close to discussing this loss is section 3. But here again the discussion is shallow.
The loss optimized in DQN is an L2 loss. It's derivative induces the TD errors. Why not start by recalling these simple facts to develop the ideas of section 3? Looking at non-uniform sampling for SGD implies doing importance sampling in order to obtain gradient estimates. And, unfortunately, I believe the authors totally miss the literature on non uniform sampling in SGD (Needell et at 2014, Zhao and Zhang 2015, Alain et al. 2016, Wang et al. 2017, Katharopoulos and Fleuret 2018). In particular, PER does not exactly sample the transitions according to the TD errors. It actually samples them according to $(\delta +c)^\alpha$. And then it weights differently the contribution of each sample in the gradient computation, precisely because it mimics (approximately) an importance sampling scheme. There is nothing on this matter in section 3 and the authors seem to ignore a major part of what actually happens in PER. Also, as is common in DQN algorithms, the loss used is actually a Huber loss (so it's equivalent to L2 for small loss values) which induce a slightly different prioritization scheme of $\min(delta,1)$.

Overall, having an explicit loss would allow showing that the property illustrated in thm 1 actually holds for any L_p and L_{p+1} losses (with a few minor modifications). It could also highlight the fact that changing the loss might change to minimum of the empirical risk (L2 implies an average, not necessarily L3) and therefore the claim that "optimizing the cubic power objective [...] has faster convergence rate" should be mitigated: it's true, but convergence to what?

Also, relating the proof of theorem 2 (which appeared unnecessarily convoluted to me) to the convergence speed of SGD as developped in (Needell 2014) for instance could be a link for a clearer and more impactful contribution.

## Insufficient sample space coverage is an issue that is linked directly with online exploration

The paper claims PER has two weaknesses: outdated priorities and insufficient sample space coverage.
I'll agree with the first and argue that the second has nothing to do with PER but rather with online RL in general. If you're interacting online with an MDP environment and can't sample states wherever you wish, you are doomed to concentrate samples around the starting state(s) unless you perform some kind of filtering when filling the replay buffer. It is a core question of exploration. PER mitigates unbalance in the replay buffer by prioritizing (just as the other non-uniform sampling methods for SGD). Model-based RL allows to obtain samples everywhere and thus the comparison seems unfair.

## Up to date priorities could be better discussed

The authors claim their approach allows to sample (approximately) according to TD errors, which is true. This, however, does not guarantee that sampling according to TD errors will provide good descent directions. First, because solely changing the sampling distribution without reweighting each sample, biases the gradients in SGD. Second, because nothing guarantees that TD errors are actually a good distribution for better gradient estimates. To understand this, we need to go back to basics again concerning why sampling according to TD errors makes sense in some cases. It makes sense because the optimal importance sampling distribution for the L2 loss is proportional in each sample, to the loss' gradient. And the TD error is a surrogate (and only a surrogate) of the loss' gradient. this explains why in some (not so rare) occasions, PER does not improve on DQN. The illustration of section 3.3 could be, again, related to the question of defining good sampling distributions as for example in (Katharopoulos and Fleuret 2018).

## Limitations of Langevin dynamics

A comment: Langevin dynamics requires a non piecewise-constant density function. Interestingly, for many MDPs, the value function is piecewise constant. Take for example the deterministic, continuous action, 2D navigation problem in a box, with a reward only at the exit, then the optimal value function is piecewise constant. This is likely to lead to piecewise constant TD errors and thus zero gradients for the Langevin dynamic. In this case, the proposed surrogate sampling scheme boils down to a Metropolis-Hastings dynamic, whose good properties fully depend on the Gaussian random variable which is not really discussed in the paper.

## Section 4.1 should be clarified

Besides the notations (already mentioned, e.g. $y(s)$ should be $y(s,\theta)$) there are a lot of clumsy formulations in section 4.1.
For instance, if $v(s)=\max_a Q(s,a;\theta_t)$ then the TD error is actually a TD error for the Bellman equation defined on $v$ functions. Then why isn't the value function $v$ directly parametrized by $\theta$? What's the need to use $Q$?
Also, when $\hat{y}(s)$ is considered constant, it is left to the reader to assume it is with respect to $s$. This assumption seems rather strong and would deserve a better discussion (despite the argument provided in appendix A.5).

## Experiments are not discussed in depth

I don't understand Figure 2. If what is represented is the distribution of states in the replay buffer, why is there no mass around the starting state or around common trajectories for DQN and PER?

The comparison of the total variation metric (why not call a spade a spade?) on Figure 3,  between PER and the distribution according to a Langevin dynamic is quite unfair. Since PER only updates TD errors in states selected in minibatches, as the authors point out, the vast majority of priorities is outdated and thus these figures are not surprising. What is the point then?

The baselines made me doubt: is PER the version with a target network, Huber loss, priorities $(\delta +c)^\alpha$, and sample reweighting?

The discussion on Dyna-Value/Frequency deserves more than a wild conjecture. The authors refer to explosive of zero gradients, imbalance in the sampled states, hyperparameter sensitivity, but the discussion remains shallow as to why Dyna-TD outperforms them and seems to perform well in general.

The part on DDPG deserves a few precisions: minibatch sampling is done according to the Langevin dynamic for the critic alone? Or is there a common sampled minibatch for the actor and the critic? What's the rationale?
Also, DDPG suffers from a number of limitations concerning its convergence speed and stability. Experimenting with the much more stable TD3 or SAC (to remain in the family of approximate value iteration algorithms) would be more convincing.

The conclusion on the Highway environment puzzles me. All algorithms optimize for cumulative (discounted) reward, so the fact that episodic return is approximately equivalent between them rather illustrates that Dyna-TD does not bring a significant advantage over PER. After that, claiming that it is better because it crashes less is a very anthropomorphic interpretation whose interest I fail to see.


**Summary Of The Paper:**

This paper deals with stochastic gradient descent for approximate value iteration and relies upon the commonly admitted result (of prioritized experience replay) that sampling states from the replay buffer according to their TD error provides better gradient estimates. The main contribution of the paper consists in using a Langevin dynamic in order to simulate the distribution of TD errors over states, in order to build minibatches in model-based reinforcement learning.

**Summary Of The Review:**

I recommend rejection of the paper.
In the review above, I have done my best to give the authors detailed info on why I believe the paper in its current state is too shallow. The idea of using a Langevin dynamic to draw states according to their TD error seems like an interesting basis to me but the current paper needs more work to be accepted for publication.
To summarize my main arguments:
- Lack of connection to the litterature on MDP solving and non-uniform sampling in SGD.
- No actual justification of why PER is (or not) sound.
- Not clear enough on what the contribution is.
- The algorithmic contribution is not studied enough, both theoretically and empirically.

---

> ### Author Response · Authors · 2021-11-22
> **Author response**
>
> We would thank the reviewer for carefully reading the paper and for valuable comments.
>
> 1. "Proper grounding about section 3"
>
> We will further improve the presentation according to your suggestions. The current results in Section 3 already provide theoretical insight for error-based sampling used in PER. In observing the difficulty of analyzing it in an RL setting, we opt to analyze it from a supervised learning perspective. PER does use importance ratio to re-weight samples; however, that is because of the mismatch between an on-policy distribution and the prioritized sampling distribution. It is irrelevant to our purpose of analyzing error-based sampling distribution.
>
> In Theorem 2, we show that the cubic loss achieves an intermediate error level ($\epsilon$) faster than the square loss, which shows that the TD-error is reduced more quickly at the early stage if we use cubic loss.
>
> Regarding background in ADP. In fact, the term ADP is rarely used in ER-related works, which can be seen from many such papers we cited.
>
> 2. “Insufficient sample space coverage.”
>
> We agree with the reviewer that insufficient sample space coverage is closely related to online RL in general (i.e., exploration-exploitation trade-off). However, we would argue that it is not the case that it "has nothing to do with PER".
>
> This is because PER also collects experiences by interacting with the environment and storing them in a buffer (it is also an online algorithm). As the reviewer noted, the insufficient sample space coverage is related to general online learning. Hence it relates to PER. In particular, we show that this coverage issue significantly makes error-based sampling less advantageous, as shown in Fig. 1. Therefore PER is also facing this issue.
>
> “the sampling distribution comparison is unfair because MBRL allows to obtain samples everywhere.”
>
> A model provides the flexibility to obtain samples, and this is why we use SGLD in MBRL to overcome the second limitation. This is a design by choice, which to our understanding, is not about fair/unfair; the purpose of the figure is to verify that our method does provide a closer distribution to the ideal one (not having the two limitations).
>
> 3. “Up-to-date priorities could be better addressed.”
>
> Thank you for pointing out the nice SGD literature to us. Please note that the theoretical results of SGD algorithms are mostly in a supervised learning setting rather than for MBRL. It is not clear yet how well these results can be adapted to an RL setting. We start from the PER work (which is already empirically shown to have strong performance) and attempt to understand the limitations of the prioritized sampling method from a supervised learning perspective. We will add relevant references and discussions in the related work.
>
> 4. “Limitations of SGLD.”
>
> If the value function becomes optimal, whether it is a piecewise-constant function or not, TD error is zero (value function satisfies the Bellman equation).
>
> Therefore, the non-zero TD error only appears when we use approximations of value functions rather than using value functions themselves.
>
> 5. “Why learn Q not V?”
>
> This is because we need to act greedily w.r.t. action values. As a variant of Dyna, our algorithm uses Q values, as is also used in the original Dyna.
>
> It is definitely doable to query a model to decide which action to take as the reviewer suggested. However, this seems not wise, as this would introduce an additional model error effect when executing the behavior policy.
>
> 6. “Why no mass around start state?”
>
> There is mass around it (no mass grids should be white color). Also, please note that each grid has width $=0.1$ in subfigure (b-d), and the agent can move $0.05$ per unit. Therefore the dark color in the left-bottom already indicates many steps around the initial states (i.e., the initial states are sampled from a small area in the left-bottom).
>
> “About DDPG.” We use both actor and critic: $Q(s,\pi(s))$ where $\pi$ is the actor and $\pi(s)$ is an approximation of the greedy action, as indicated in our paper.
>
> On the autonomous driving domain, if the speed is higher, then the reward is larger. Our agent learns to sacrifice some speed to reduce crashes, which means safety in practice.
>
> 7. Section 4.2 is the empirical verification to show our method provides a better sampling distribution than vanilla PER by addressing the two limitations. As a result, we put such empirical results immediately after explaining the sampling method. Section 5 contains empirical results in terms of learning curves, not directly about sampling distributions.

---

> > ### Comment · Reviewer_VdDV · 2021-11-28
> > **Feedback**
> >
> > I have read the authors' responses.
> >
> > I think there may be a lack of hindsight both in the paper and the authors' response (no offense meant, I strongly encourage strengthening the analysis in future work). I also feel the authors dodge a number of important questions. A few examples below.
> > Both DQN or DDPG are approximate value iteration methods: they are a sequence of supervised learning problems aimed at finding Q*. So the supervised learning literature is *very* relevant to this work (and not marginally).
> > I don't think the fact that ADP is not quoted so often in the literature should be an argument in a scientific paper: most RL work today stems from AVI (approximate value iteration) and in the precise case of this paper, I don't see how looking at convergence results can be decoupled from the well-studied convergence bounds of ADP.
> > Also, biasing a sampling distribution to evaluate an expectation will bias the result, hence I fail to see how one can write that re-weighting is irrelevant to the purpose of analyzing error-based sampling distribution.
> > The answer about limitations of Langevin dynamics doesn't address the question.
> >
> > Overall, I still believe the contribution is not mature and strong enough for publication. I keep my score.

---

### Official Review · Reviewer_iRvC · 2021-11-02

**Correctness:** 3
**Technical Novelty And Significance:** 3
**Empirical Novelty And Significance:** 3
**Recommendation:** 5
**Confidence:** 4

**Main Review:**

### Strengths

- This paper is well-written. The motivation and core mechanism behind their method is clearly and concisely explained, and ample, additional details are provided in the appendix.
- The experimental results are presented in a simple and digestable manner. They highlight gains with respect to the key baseline, which is simply standard PER. These results are convincing, as their method both improves on PER in terms of both returns and distance to the ground-truth PER distribution (if all transitions were updated rather than just those in the minibatch).
- The experimental setting is diverse: It contains both a gridworld environment, standard continuous control Gym tasks, and a driving task based on driving around a roundabout.
- In sum, their problem motivation, method motivation and description, and experimental results make the case for Dyna-TD quite convincing.

### Weaknesses

- While the paper looks at the number of SGLD updates n in {10, 30} and show that larger n does not make a large difference in accuracy to the ground-truth PER distribution, it seems like a missed opportunity to further decrease n to see just how few updates are needed to retain the observed performance and accuracy gains.
- Related to the above point, it would make the method more convincing if the authors could show that the SGLD updates do tend to approximately converge after just n=10 updates.
- A key missing baseline in their RL experiments is Full-PrioritizedER. This baseline is required to make sure that Dyna-TD is working for the hypothesized reasons. I believe this may be as simple as adding their MountainCar result in Figure 1 (c) to Figure 4.
- Their method, while called Dyna-TD, diverges from Dyna in that the agent only trains on samples collected from the experience replay buffers, which in this case, acts as a purely memory-based model. Therefore, it seems that an algorithm that better matches the name "Dyna-TD" would also train the agent on experiences collected online in each update cycle, as done in Dyna. In contrast, their method seems more accurately described as a form of PER, e.g. Langevin-Dynamics PER. Since the authors make a considerable effort drawing the connection between ER and Dyna, it would benefit the paper to benchmark a version of HC-Dyna that mirrors Dyna in also performing updates on transitions collected online.
- Several interesting empirical findings warrant deeper analysis: For example,
    - i) Why does Dyna-TD result in fewer car crashes and lower average speed despite attaining similar asymptotic episodic returns during training?
    - ii) Why does Dyna-Value and Dyna-Frequency perform worse? The authors put forth a hypothesis that perhaps these variants bias toward transitions with low TD-error, leading to slower learning, but do not verify this hypothesis with data.
- Given the similarity of this work to Pan et al, 2020, which introduced HC-Dyna and Dyna-Frequency, I believe having more in-depth analysis that highlights why the new method, Dyna-TD works better than Dyna-Value and Dyna-Frequency, is important for making this paper a more substantial, standalone contribution.
- Dyna-TD introduces two hyperparameters, the number of SGLD updates n and the variance of the noise variable X. The paper would benefit from experiments showing how X impacts the performance of Dyna-TD.
- The authors should make clear in the paper that their method, Dyna-TD, only applies to continuous state-space environments.

**Summary Of The Paper:**

This paper proposes an alternative method for performing Prioritized Experience Replay (PER), which avoids issues of inadequate state coverage and staleness in priority scores. Their method is based on a result from stochastic langevin dynamics, which shows their specific stochastic gradient Langevin dynamics (SGLD) update to online states leads to the ground-truth PER distribution in the limit. Updating collected states this way and adding them to an additional "stochastic-control queue" from which sampled minibatch transitions are mixed with those sampled from a standard experience replay buffer. Empirically, this paper shows this SGLD-based method leads to a closer approximation to the ideal prioritized experience replay distribution over recent transitions than standard PER.

**Summary Of The Review:**

The paper provides strong motivation and empirical results supporting their new method Dyna-TD.  However, given their method is a simple extension of HC-Dyna and Dyna-Frequency from Pan et al, 2020, where they replace the hill-climbing objective with a TD-error objective, it seems important to include more analysis comparing and contrasting TD-Dyna to HC-Dyna, as well as a deeper understanding of Dyna-TD's learning dynamics. For these reasons, while I find the method itself promising, I recommend this paper as being marginally below the acceptance threshold.

---

> ### Author Response · Authors · 2021-11-22
> **Author response**
>
> We thank the reviewer for carefully reading our paper and for helpful feedback. We address the main concerns as follows.
>
> 1. and 2. "Small $n$"
>
> Thank you for raising this point, and we will add results for small $n$ values.
>
> Please allow us to explain why using a larger n likely makes more sense. Recall that PER has two limitations: 1) sample space coverage; and 2) outdated priorities. First, with a large $n$, at least those states in the buffer would get sufficiently updated. Second, PER only updates the priorities in the sampled mini-batch. Hence, using a large n (in PER, $n$ is the number of mini-batch updates per environment time step) makes more samples have update-to-date priorities.
>
> 3. "Full-PrioritizedER"
>
> We agree with the reviewer this is a reasonable point. However, unfortunately, doing Full-PrioritizedER is infeasible in a larger domain. Typically $d$, as the number of data points is increasing all the time. And note that Fig. 1(c) uses a relatively small buffer for the purpose of comparison, which should not be a reasonable implementation choice for larger experiments.
>
> 4. "Dyna-TD vs. Dyna"
>
> Sorry for the confusion. We would like to note that our method does use trajectories collected from interacting with the environment. For example, consider the "Add $(s_t, a_t, s_{t+1}, r_t)$ to $B_\text{sr}$" line in Alg. 1, both $s_{t+1}$ and $r_t$ are from the environment. Therefore, our method does "also train the agent on experiences collected online in each update cycle, as done in Dyna".
>
> 5. "empirical findings warrant deeper analysis"
>
> i) In this task, if the speed is higher, then the reward is larger. Then it is evident that our agent learns to sacrifice some speed to reduce crashes.
>
> ii) For Dyna-Value, it would frequently sample high-value states, even if it learns well enough on those states. Similarly, Dyna-Frequency would keep sampling high-frequency states, even if it learns the value function well for these states. In contrast, in this case, the Dyna-TD would query some other states that are currently not well-learned, which is arguably a more reasonable choice not to repeatedly sample well-learned states.
>
> Another issue for Dyna-Frequency is it requires calculating higher-order (second or third) derivatives, which precludes using several activations such as ReLU. Empirically, we find the gradient w.r.t. $s$ in Dyna-Frequency have very different numerical scales on different domains, making it quite challenging to select hyper-parameters. It is not clear and remains to be better understood the deeper reasons.
>
> 6. "Dyna-TD introduces two hyperparameters"
>
> For the number of updates $n$ in SGLD, in Fig. 3, Dyna-TD-Long is the version of using a large $n$, which yields a closer sampling distribution to the ideal one than the Dyna-TD. While in the experimental section, we show that it is actually reasonably good even if we use a relatively smaller $n$. We will report more results about the variance of the noise variable $X$. Thank you for your suggestions.
>
> 7. "make clear in the paper that their method, Dyna-TD, only applies to continuous state-space environments"
>
> Thank you, and we will make this point clear as suggested.

---

> > ### Comment · Reviewer_iRvC · 2021-11-24
> > **Thanks for the response**
> >
> > I appreciate the authors' responses. I still believe that the driving environment results warrant a more in-depth treatment to justify the "improvement" resulting from their method. Otherwise, it feels a bit like moving the goal posts.
> >
> > Could you say anything about the convergence of your method's SGLD to the true PER distribution? It seems that the PER distribution should be constantly changing during training, and SGLD converges to this distribution in the limit. Considering that each update cycle only performs a finite number of SGLD updates, it is unclear to me why we can rely on this algorithm accurately approximating the true PER distribution, when this distribution is non-stationary over the course of training.

---

> > > ### Author Response · Authors · 2021-11-25
> > > **Thanks for your reply.**
> > >
> > > Thanks for the insightful comments.
> > >
> > > You are right. We may not be able to run SGLD for too many steps. However, we can still rely on it because we expect it provides a closer distribution to the desired one than conventional PER. This is the reason why we want to validate the sampling distribution acquired by our SGLD algorithm empirically. Figure 3(a)(b) shows the results with two different weighting schemes when computing the distance between distributions. As one can see, running long time steps (Dyna-TD-Long) in SGLD does provide a closer distance to the desired sampling distribution. But Dyna-TD---which only runs a small number of steps---still provides a distribution better than PER does.

---

### Official Review · Reviewer_DyBg · 2021-11-03

**Correctness:** 3
**Technical Novelty And Significance:** 2
**Empirical Novelty And Significance:** 2
**Recommendation:** 3
**Confidence:** 4

**Main Review:**

The paper proposes to use prioritized sampling in conjunction with a model-based RL algorithm and attempts to solve issues with PER using up-to-date priorities using SGLD and using a dynamics model to augment data to cover the space. The paper is interesting, but there are several issues in my opinion:

- From the intro: "The idea behind prioritized sweeping is quite intuitive: we should give high priority to states whose absolute TD errors are large because they are likely to cause the most change in value estimates." -- this is not intuitive to me, controlling TD errors on the training distribution may not give rise to a better performing policy!

- Theorems 1 and 2 apply only to the supervised learning setting or FQI, but that is not what we do in practice. We typically update the target network halfway. Is prioritization still always good in this case, and improves convergence rate? To me, this is not clear -- and I think the answer is no. Let's say that the Q-function is diverging, then prioritization will cause the network to fit diverged target values compared to the more stable ones as they have higher TD errors. So I don't think Theorems 1 and 2 really reflect reality.

- I don't buy the algorithm proposed. When doing SGLD to find states with high TD error, one can find arbitrary far away states. Augmenting them with model data to generate target values and then training on them may actually exacerbate the problem since the model is going to be inaccurate and may predict very high or incorrect reward values. These problems do actually arise in model-based offline RL, even though the states are obtained by unrolling short model rollouts under the learned model, on domains where models can be fit well (e.g., continuous control gym benchmarks). So I am unsure if the technique proposed is generally useful, unless we control for OOD errors. it might work well on the toy domains in the experiments, but may not still be general.

- The results on continuous control tasks are not convincing. First of all, they use DDPG, which is known to be a bad algorithm. If you can show this on TD3 or SAC, I would be somewhat more convinced. Also what if you try the algorithm on a domain where the model is slightly inaccurate, or maybe do an ablation for that? It is good to know what happens when the learned model is inaccurate.

**Summary Of The Paper:**

The paper proposes to use dynamics moel to augment data to alleviate some issues with PER pertaining to insufficient coverage, and outdated priorities. Some empirical validation shows that the algorithm performs well on some toy control tasks.

**Summary Of The Review:**

I feel like the paper does not indicate (1) why we should build on PER (2) in my opinion, the algorithm is flawed and may not scale otherwise to problems where dynamics models are inaccurate (3) the empirical results are not convincing. So, I am going for a reject.

---

> ### Author Response · Authors · 2021-11-22
> **Author response**
>
> We thank the reviewer for careful reading and helpful feedback. We address the main concerns as follows.
>
> 1. "controlling TD errors on the training distribution may not give rise to a better performing policy."
>
> We agree with the reviewer, and we do not claim that controlling TD errors gives rise to a better-performing policy. What we wanted to say here is to provide an intuitive explanation of doing prioritized sweeping (from literature, Moore and Atkeson, 1993), i.e., once a transition causes a big change, the predecessor state leading to that state should be worth learning. As prioritized sweeping is a widely used method in RL, its intuition is also worth considering.
>
> 2. "Theorems 1 and 2 ... not what we do in practice"
>
> We acknowledge this gap between the theory and the practice, and we did not claim our results have no gap to the practice.
>
> The theorems provide intuitions about the PER, rather than accurately characterizing the practical cases (which is, of course, a valuable but very difficult question).
>
> On the other side, the example suggested by the reviewer could also not deny the intuitions of our theorems, since, in practice, divergence is not always the case (but we agree that in some cases, this would happen). At least, our theorems provide valid intuitions in some cases (but not all cases as noted by the reviewer).
>
> 3. "one can find arbitrary far away states"
>
> The pitch of this paper is about designing error-based sampling distribution. It is worths another work to study model learning techniques by taking into account simulated experiences sampling.
>
> In fact, as the initial work to mitigate the issues of PER, we already did a reasonable amount of work to show our algorithm's performance with an online learned model. We provide the learning curves of Dyna-TD with an online learned model on all benchmark domains, Mujoco domains, and an autonomous driving domain.
>
> 4. "The results on continuous control tasks are not convincing"
>
> We consider DDPG as a reasonable choice of baseline since it inspired many follow-ups and still remains to be a widely used RL methods (one can argue its performance sometimes is not satisfactory, but it provides a reasonable objective for comparison). We thank the reviewer for suggesting conducting experiments using more baseline methods.
>
> Moore, A. W. and Atkeson, C. G. Prioritized sweeping: Reinforcement learning with less data and less time. Machine learning, pp. 103–130, 1993.

---

> > ### Comment · Reviewer_DyBg · 2021-11-23
> > **Reviewer Response**
> >
> > Thanks for the response, however, my points are still not largely addressed. I feel like the authors' response primarily dodges the central questions I asked, and I would still request authors to address the points I mention below.
> >
> > First of all, I would like to point out the overclaim: "In fact, as the initial work to mitigate the issues of PER, we already did a reasonable amount of work" --> this is not the first work to mitigate issues in PER, many works have been proposed to improve prioritized sampling schemes either building on PER or coming up with a different scheme.
> >
> > I still don't think the proposed method performs well. The improvements on DDPG are minimal and worse than the default performance of TD3 and SAC, which makes it completely unconvincing. Can you please some experiments with TD3 and SAC? Atleast on some gym domains? It does make a big difference in my experience, and so I am just not sure of your method unless I see results with more advanced methods.
> >
> > Regarding values under a model -- I don't buy this without an analysis, could you do an empirical analysis on some toy gridworld domains understand the errors in the priorities and the SGLD sampling scheme, and compare your practical method to an oracle version of the method?

---

> > > ### Author Response · Authors · 2021-11-24
> > > **Thanks for your response.**
> > >
> > > Thanks a lot for your quick response. We do appreciate your suggestions.
> > >
> > > 1. Sorry for the overclaim.
> > >
> > > 2. Sure, we will also test these algorithms in our framework.
> > >
> > > 3. Sorry for the confusion and our unclear description. If we understand correctly, we should have already done what you suggested. Please look at Figure 3(a)(b). The "desired distribution" is the one computing the prioritized sampling distribution with complete environment knowledge (i.e., true model, state-space, etc.) without using SGLD. The dashed black line uses an online learned model with SGLD, and the solid black line uses the true model with SGLD. As you can see, though the dashed line is indeed farther away from the desired distribution than the solid line, it is still better than the vanilla PER. We agree that in a highly complex environment, a special model learning method and heavy engineering may be necessary. It should be worth a separate work.
> > >
> > > We are not fighting for acceptance. We are just seriously considering your suggestion to find the best way to improve our work.

---

### Official Review · Reviewer_xiwV · 2021-11-06

**Correctness:** 2
**Technical Novelty And Significance:** 2
**Empirical Novelty And Significance:** 2
**Recommendation:** 3
**Confidence:** 4

**Main Review:**

The idea in the paper is interesting. However, I can't follow some discussions and am not convinced by the arguments from the author.

1. Theorem 3 basically says that cubic loss can learn faster than quadratic loss in some scenarios. However, different loss function might allow different learning rate.
2. Why would we sample $s$​ in the full state space? There are many states that our current policy doesn't access, so why should we care about the TD loss for these states? As shown in Figure 2, PER doesn't waste its time reducing the TD loss of black cells, while Dyna-TD does so.
3. The outdated priority issue is not solved, either. After updating value on a batch, the priorities also changes, so the search control queue $B_{sc}$​ is outdated again.
4. SGLD in Section 4.1: Please explain what the temperature parameter is. In addition, please make it clear that the stationary distribution of $s$ depends on the the variance of $X_i$.
5. The importance ratio, which is an important aspect of PER, is not discussed in this paper. With importance ratio, the loss is unchanged, so Section 3.1 becomes useless. In Figure 1, is the importance ratio added? If not, the comparison doesn't make much sense as we're comparing the test error of two algorithms with different training error.
6. I don't see the point of being model-based. In Algorithm 3, how is the learned dynamics model used? Is it used to compute $\hat y$? Please elaborate.
7. In continuous control tasks, how is the max in Equation 3 computed?
8. The theory on Section 3.1 is kind of simple: Theorem 1 can be simply viewed as importance sampling. Theorem 2 is basically univariate regression with mse/cubic loss.

**Summary Of The Paper:**

This paper takes a deep look at prioritized experience replay, a popular technique in deep reinforcement learning. The paper gives insights on why error-based prioritized experience replay can help when the importance ratio is unused. This paper also pointed out two limitations of prioritized experience replay, which are outdated priorities and insufficient coverage of state space, and the author proposed to use SGLD to solve the limitations. Experiments show that the proposed method leads to a good coverage of state space and improve the return of the training algorithm.

**Summary Of The Review:**

I feel like there is a plenty room of improvement. Some key aspects of PER is not discussed in the paper, I'm not convinced the proposed solution solved the issue, and the theory is not very technical either. Thus I don't recommend acceptance.

---

> ### Author Response · Authors · 2021-11-22
> **Author response**
>
> We thank the reviewer for carefully reading our paper and for valuable feedback. We address the main concerns as follows.
>
> 1. “different loss functions might allow different learning rate.”
>
> Our results extend to using different learning rates for different losses. Consider using $\eta_1$ for the square loss and $\eta_2$ for the cubic loss. Then we have, Eq. (24) holds with $\eta$ replaced with $\eta_1$, and Eq. (32) holds with $\eta$ replaced with $\eta_2$. As a result, Eq. (33) becomes $\frac{1}{ \eta_1} \cdot \log{\left( \frac{\delta_0(i)}{\epsilon(i)} \right)} - \frac{1}{ \eta_2 } \cdot \left( \frac{1}{ \epsilon(i)} - \frac{1}{\tilde{\delta}_0(i) } \right) $. Then depending on whether $\eta_1 > \eta_2$ or $\eta_1 \le \eta_2$, similar results hold: with $\delta_0(i)$ larger than some initial values (which is a function of $\eta_1$ and $\eta_2$), cubic loss is faster than square loss for achieving a smaller threshold value $\epsilon(i)$.
>
> 2. "Why sample s from the full state space? PER does not waste samples … "
>
> As shown in the theoretical (Thm. 1) and empirical evidence (Fig. 1), sufficient sample space coverage is an important factor for prioritized sampling to show clear benefits. In the experiment section, our results show that our algorithm outperforms PER.
>
> 3. “Outdated priority issue is not solved … ”
>
> At each time step, SGLD is sampling states using the current parameter, and thus all the simulated experiences are using the updated priorities.
>
> Therefore, rather than taking actions in the environment and refreshing the real priorities in the PER buffer, we use a model to get simulated priorities, which can be calculated more efficiently.
>
> 4. Sorry for the missing details. Let the $\sigma^2$ be the variance of the r.v. X_i, then the temperature parameter in the Gibbs distribution is $2\alpha_h/\sigma^2$.
>
> Thank you for pointing this. We will make it clear that the stationary distribution of $s$ depends on the variance of $X_i$.
>
> 5. Please note that the original PER paper raises the importance ratio to the power of $\beta$, which is annealing from $0$ (at the beginning) to $1$ (asymptotically), cf. Sec. 3.4 of the PER paper (Schaul et al., 2016).
>
> Therefore, during early learning, $\beta$ is close to zero (strictly smaller than $1$), which is equivalent to using a higher power loss (larger than $2$). This point has also been confirmed by a concurrent work (Fujimoto et al., 2020 in the reference; their Sec. 5.1 and Thm. 3). Given this, a small $\beta$ value at the beginning is consistent with our Thm. 2, i.e., explaining the theoretical insight of error-based sampling during the early learning stage.
>
> Thank you for raising this point, and we will add the above clarifications into the subsequent versions.
>
> 6. Sorry for the missing details. There are two main reasons for being model-based: sampling and learning.
>
> 1) Sample. Yes, we need to estimate the bootstrap target $\hat{y}$ in the SGLD sampling process;
>
> 2) Learning. Note that we only have stated in the search-control queue. Once we pair them with actions, then we get state-action pairs. We need to query the model to sample the next state and reward (please see Algorithm 1, the "Sample $s^\prime, r \sim \mathcal{P}(s,a)$" step).
>
> 7. In continuous setting, a policy network is learned to output one action given a state, i.e., $\pi(s)$ is an approximation of $arg max_{a} Q^{\pi}(s, a)$, and thus $Q(s, \pi(s))$ is an approximation of $\max_{a} Q^{\pi}(s, a)$, which is a standard configuration in continuous setting (e.g., see Silver et al., 2014, "Deterministic Policy Gradient Algorithms").
>
> 8. We would politely argue that a theory is simple does not mean that it cannot provide useful insights to our understanding. A simple and useful theorem should be considered as a strength.
>
> Our theory shows the connection between PER and cubic loss, and it also reveals that cubic learns faster in the early stage, which to our knowledge, was not known before to explain prioritized sampling.
>
> We would like to thank the reviewer again for the valuable comments. Any further suggestions would be highly appreciated.

---

### Author Response · Authors · 2021-11-23
**General Response**

We sincerely appreciate all the reviews.

We responded to all reviewers. Irrespective of the decision on the paper, we would appreciate it if the reviewers could indicate any further concerns (if any) to help us better improve the paper.

---

### Decision · Program_Chairs · 2022-01-20

**Decision:**

Reject

**Comment:**

This work provides a theoretical analysis of Prioritized Experience Replay (PER ) in a supervised learning setting, points out limitations of PER and proposes a model-based approach to address these shortcomings for continuous control problems.

Strengths:
-----------
The overall problem was motivated well
Reviewers agree that this proposed algorithm has promise
Overall the paper is well written
a diverse set of experiments is provided

Weaknesses:
---------------
reviewers point out some clarity issues
The theoretical analysis is performed in a supervised learning setting, and it is unclear how the resulting analysis transfers to the RL setting
There are some concerns (theoretical/technical) wrt to the proposed algorithm.
The analysis of the experiments is lacking in depth. For instance, no analysis of why the proposed algorithm outperforms very related baselines. Furthermore, it's unclear why for the autonomous driving experiment the algorithms achieve the same return, but the proposed method leads to less crashes.

Rebuttal:
----------
The authors have addressed many of the clarity issues. However, I agree with the reviewers theoretical concerns and deeper analysis requests were not addressed in a significant manner.

Summary:
------------
Overall this manuscript investigates an important problem and provides a promising algorithm. However, some theoretical/technical concerns remain and a deeper analysis of results is required. Hence my recommendation is that in it's current form the manuscript is not quite ready yet for publication.